# Assessment of Regional Aerosol Radiative Effects under SWAAMI Campaign – PART 1: Quality-enhanced Estimation of Columnar Aerosol Extinction and Absorption Over the Indian Subcontinent

Harshavardhana Sunil Pathak[1], Sreedharan Krishnakumari Satheesh[1,2], Ravi Shankar Nanjundiah[1,2,3], Krishnaswamy Krishna Moorthy[1], Sivaramakrishnan Lakshmivarahan[4], and Surendran Nair Suresh Babu[5]

[1]Centre for Atmospheric and Oceanic Sciences, Indian Institute of Science, Bangalore, India
[2]Divecha Centre for Climate Change, Indian Institute of Science, Bangalore, India
[3]Indian Institute of Tropical Meteorology, Pune 411008
[4]School of Computer Science, University of Oklohama, Norman, United States of America
[5]Space Physics Laboratory, Vikram Sarabhai Space Centre, Thiruvananthapuram, India

**Correspondence:** Harshavardhana Sunil Pathak (rhsp19@gmail.com)

**Abstract.** Improving the accuracy of regional aerosol climate impact assessment calls for improvement in the accuracy of regional aerosol radiative effects (ARE) estimation. One of the most important means of achieving this is to use spatially homogeneous and temporally continuous datasets of critical aerosol properties, such as spectral aerosol optical depth (AOD) and single scattering albedo (SSA), which are the most important parameters for estimating aerosol radiative effects. However, observations do not provide the above; the space-borne observations though provide wide spatial coverage, are temporally snap shots and suffer from possible sensor degradation over extended periods. On the other hand, the ground-based measurements provide more accurate and temporally continuous data, but are spatially near-point observations. Realizing the need for spatially homogeneous and temporally continuous datasets on one hand and the near-non-existence of such data over the south Asian region (which is one of the regions where aerosols show large heterogeneity in most of their properties), construction of accurate gridded aerosol products by synthesizing the long-term space-borne and ground-based data, has been taken up as an important objective of the South West Asian Aerosol Monsoon Interactions (SWAAMI), a joint Indo-UK field campaign, aiming at characterizing aerosol-monsoon links and their variabilities over the Indian region.

In the Part-1 of this two-part paper, we present spatially homogeneous gridded datasets of Aerosol Optical Depth (AOD) and Absorption Aerosol Optical Depth (AAOD), generated for the first time over this region. These data products are developed by merging the highly accurate aerosol measurements from the dense networks of 44 (for AOD) and 34 (for AAOD) ground-based observatories of Aerosol Radiative Forcing NETwork (ARFINET) and AErosol RObotic NETwork (AERONET) spread across the Indian region, with satellite-retrieved AOD and AAOD, following statistical assimilation schemes. The satellite data used for AOD assimilation includes AODs retrieved from MODerate Imaging Spectroradiometer (MODIS) and Multiangle Imaging SpectroRadiometer (MISR) over the same domain. For AAOD, the ground-based Black Carbon (BC) mass concentration measurements from the network of 34 ARFINET observatories and satellite-based (Kalpana-1, INSAT-3A) infrared (IR) radiance measurements, are blended with gridded AAODs (500 nm, monthly mean) derived from Ozone Monitoring Instrument

(OMI)-retrieved AAODs (at 354 nm and 388 nm). The details of the assimilation methods and the gridded datasets generated are presented in this paper.

The merged, gridded AOD and AAOD products thus generated, are validated against the data from independent ground-based observatories, which were not used for the assimilation process, but are representative of different subregions of the complex domain. This validation exercise revealed that the independent ground-based measurements are better confirmed by merged datasets than the respective satellite products. As ensured by assimilation techniques employed, the uncertainties in merged AODs and AAODs are significantly less than those in corresponding satellite products. These merged products also exhibit all important, large-scale spatial and temporal features which are already reported for this region. Nonetheless, the merged AODs and AAODs are significantly different in magnitude, from the respective satellite products. On the background of above mentioned quality enhancements demonstrated by merged products, we have employed them for deriving the columnar SSA and analysed its spatio-temporal characteristics. The columnar SSA thus derived has demonstrated distinct seasonal variation, over various representative subregions of the study domain. The uncertainties in the derived SSA are observed to be substantially less than those in OMI SSA. On the backdrop of these benefits, the merged datasets are employed for the estimation of regional aerosol radiative effects (direct), the results of which would be presented in a companion paper; Part-2 of this two-part paper.

## 1    Introduction

The climate forcing potential of atmospheric aerosols is well accepted by the global scientific community and policy makers [*IPCC, 2013*]. This forcing can affect Earth's hydrological cycle (Ramanathan et al., 2001; Bollasina et al., 2011), increase the stability of the atmosphere (Ramanathan and Carmichael, 2008; Jacobson and Kaufman, 2006; Petäjä et al., 2016) and can have significant impact on Indian summer monsoon (Lau and Kim, 2006). Along with these climatic impacts, aerosols are shown to have an adverse effects on human health (Dockery et al., 1993; Seaton et al., 1995; Pope III et al., 2002). Accurate assessment of these impacts still remains a challenge, primarily due to the inadequate spatio-temporal coverage of the aerosol properties such as the Aerosol Optical Depth (AOD) and Single Scattering Albedo (SSA), and the large uncertainties prevailing in the available database. This is especially so over the Indian region, which is among regions having high aerosol loading, that shows large heterogeneities in its spatial and temporal characteristics. These heterogeneities are primarily because of wide diversity in geography, anthropogenic activities and meteorological features at meso-scale and synoptic scale. As demonstrated by the past studies (Haywood and Shine, 1995, 1997; Heintzenberg and Helas, 1997; Russell et al., 2002; Takemura et al., 2002; Loeb and Su, 2010; Babu et al., 2016), a small change in the SSA can even alter the sign of aerosol radiative forcing (at top of atmosphere) from positive (warming) to negative (cooling), especially over highly reflecting surfaces. The large spatial heterogeneity in the surface reflectance of the land-mass over this region, and its seasonality makes the aerosol radiative forcing estimation all the more complex. Given this background, construction of gridded datasets of aerosol parameters, especially AOD and SSA, with reduced uncertainties and fairly homogeneous spatial and temporal distribution over the region, becomes imperative. One way to achieve this is data assimilation, a mathematical technique of generating a dataset with reduced uncertainties by

systematically combining multiple datasets (which individually may have higher uncertainties) (Kalnay, 2003; Lewis et al., 2006).

Various space-borne sensors aboard remote sensing satellites (such as MODIS, MISR, OMI etc) provide the global datasets for spatial and temporal distributions of AOD and Absorption AOD (AAOD) (Kaufman et al., 1997; Diner et al., 1998; Chu et al., 2002; Remer et al., 2005; Torres et al., 2007). Despite their wide spatial coverage, the satellite retrieved data suffers from substantial biases and uncertainties due to cloud contamination, various assumptions made during the retrieval procedure, large spatial heterogeneity in the ground reflectance (over the heterogeneous landmass), and also due to very little information on variation during a day (due to snapshot nature of measurements). In addition, satellite retrievals suffer from issues regarding sensor calibration (Zhang and Reid, 2006; Jethva et al., 2014). Especially over the land with heterogeneous surface reflectance, satellite-retrieved AODs depict higher uncertainties (Jethva et al., 2009). On the other hand, being direct measurements, AOD or Black Carbon (henceforth BC which is the primary absorbing aerosol specie) mass concentration measured respectively using ground-based, periodically calibrated sun photometers and Aethalometers, are quite accurate, have large temporal coverage in a day as well as over the years (Moorthy et al., 1989; Holben et al., 1998; Hansen and Novakov, 1990; Babu et al., 2004) along with smaller uncertainties than their satellite counterparts. However, their limited spatial representativeness (more like point measurements) calls for a dense network of observations for a reasonable spatial coverage; even then remote and inaccessible areas remain under-sampled. Moreover, practical constraints result in spatially non-uniform distribution of the ground-based stations.

These limitations of satellite retrieved (SR) and ground (GR) measured aerosol parameters restrict their applicability for climate impact assessment studies over heterogeneous regions, like the vast Indian region. However, the relative advantages of these two datasets could be effectively employed for improving regional radiative forcing estimation if these different independent datasets could be assimilated to generate a more accurate and spatio-temporally continuous gridded dataset following established statistical assimilation techniques.

There have been a few efforts in the past to combine AODs from various sources, regionally and globally. Collins et al. (2001) have assimilated AODs retrieved by Advanced Very High Resolution Radiometer (AVHRR) with those simulated by Multi-scale Atmospheric Transport and CHemistry (MATCH) model for generating forecasts of aerosols during INDian Ocean EXperiemt (INDOEX). Since then, a few more studies have focused on assimilating satellite retrieved aerosol products with those simulated by regional/global chemistry transport models (Yu et al., 2003; Generoso et al., 2007; Niu et al., 2008; Zhang et al., 2008). Benedetti et al. (2009) have incorporated AOD assimilation as an integral part of weather forecasting system at European Centre for Medium-Range Weather Forecasts (ECMWF). However, in all these efforts, the focus was to use satellite products with chemistry transport models and none of these studies has assimilated AOD observations from ground-based network of sun-photometers with the corresponding satellite retrieved parameters. Probably the first effort in this direction was by Chung et al. (2005) who have assimilated monthly mean AODs retrieved by MODerate Imaging Spectroradiometer (MODIS) with those simulated by global chemistry transport model and the resulting AODs are further integrated with monthly mean AOD measurements from Aerosol RObotic NETwork (AERONET) in order to generate the global merged AOD product. Over the Asian region, Adhikary et al. (2008) have assimilated monthly mean AERONET AODs with monthly averaged

MODIS AODs and these combined AODs are further assimilated with those simulated by regional chemistry transport model. Nonetheless, over the Indian region (bounded between 0.5° N–34.5° N and 65.5° E–96.5° E, figure 1), both of these studies have employed ground-based measurements from just two AERONET stations (Kanpur 26.51° N, 80.23° E and Hanimadhoo 6.74° N, 73.17° E). Due to this, the final assimilated AODs (for Chung et al. (2005) and Adhikary et al. (2008)) over most parts of Indian region are largely represented by satellite retrieved AODs with their inherent large uncertainties as discussed earlier. More recently, Singh et al. (2017) have combined AODs simulated by ECMWF with those retrieved by MODIS, Multiangle Imaging SpectroRadiometer (MISR) as well as in situ measured AODs by total 35 AERONET stations spread over Indian as well as Arabian region. However, even in this case employing about 17 AEORNET stations over Indian region, most of these stations were in the monsoon trough region and north-east India with no representation of other parts of the domain. The situation is still worse for SSA.

Thus, developing spatially and temporally continuous gridded products for AOD and SSA using long-term measurements from the dense network of ground based aerosol observatories (covering most parts of the Indian region) and satellite retrieved products still remained a dire necessity. This was recognised as one of the most important objectives of the South West Asian Aerosols Monsoon Interactions (SWAAMI) (https://gtr.ukri.org/projects?ref=NE%2FL013886%2F1) (Morgan et al., 2016), a co-ordinated field campaign undertaken jointly by the Indian and the UK scientists, and formed its important package.

Accordingly, we have used long-term (2001–2013) measurements of AOD at 550 nm from the two widely used space-borne sensors, MODIS and MISR, over the Indian region and the accurate, quality checked AOD from a network of 44 ground-based sun-photometers (ARFINET and AERONET) for the same period to generate a gridded dataset for AOD using a modified form of a well established data assimilation technique. On the similar lines, we have also generated a spatially homogeneous gridded product for absorption AOD (AAOD), by combining the AAODs estimated using ground-based BC measurements and space-borne infrared radiance measurements (to delineate the dust contribution to AAOD) with AAODs (500 nm) derived from Ozone Monitoring Instrument (OMI) retrievals. These merged datasets for AAOD and AAOD are further employed to estimate columnar SSA at $1° \times 1°$ over the domain.

In part–1 of this 2-part paper we provide the details of datasets employed for merging and the assimilation methodologies used (for the merging process), in section 2 and 3 respectively. The validation of merged AODs and AAODs against an independent ground-based measurements, which did not take part in assimilation, is presented in section 4.1. The merged data sets are then used to examine the spatial distribution of AOD and AAOD over the Indian region (section 4.2) and to delineate their seasonality over the spatially homogeneous sub-regions of the study domain (section 4.2 and 4.3). Further, using the validated, merged datasets, the gridded product for columnar SSA is derived and the regional as well as sub-regional scale SSA characteristics are presented in section 4.3.

## 2 Database

### 2.1 Satellite Retrieved (SR) AOD

Monthly mean AOD (at 550nm) product from MODIS on board Aqua and Terra satellites (Kaufman et al., 1997) (L3, collection 6, https://modis.gsfc.nasa.gov/data/), as well as from MISR on board Terra satellite (Diner et al., 1998) (L3, https://misr.jpl.nasa.gov/), are used as the background data in this study. The MODIS AOD product having spatial resolution of $1° \times 1°$, is constructed by merging AODs retrieved with enhanced Deep blue algorithm (Hsu et al., 2013; Sayer et al., 2013) and Dark Target algorithm (Levy et al., 2013), in order to provide AODs over bright surfaces (deserts, arid regions, semiarid regions etc.) as well as oceans. The MISR AOD product at 555 nm, with spatial resolution of $0.5° \times 0.5°$ is re-gridded to $1° \times 1°$ resolution for combining with MODIS AODs for minimizing data gaps in the background AODs due to non-availability of MODIS data. Being derived from finer resolution measurements (1 km), MODIS AOD product is preferred over MISR AOD while constructing background data. In the present study, AODs from MODIS-Terra formed a first layer of background data with gaps in it being filled by AODs from MODIS-Aqua, if present. Any data gaps existing further were filled with the re-gridded MISR AODs. Henceforth, the word SR AOD will refer to this integrated satellite retrieved AOD (MODIS + MISR).

### 2.2 Satellite Retrieved Absorption AOD

This is obtained from the Ozone Monitoring Instrument (OMI) on-board AURA satellite, which measures the upwelling radiations in the wavelength range of 270–500 nm, at the top of the atmosphere (Levelt et al., 2006). UV aerosol index, AOD and AAOD at 354 and 388 nm are then derived by incorporating the measured backscattered radiation into the inversion algorithm (OMAERUV), which makes use of pre-computed reflectance by a set of aerosol models, as detailed by Levelt et al. (2006) and Torres et al. (2005). The AOD and AAODs are then extrapolated to 500 nm by considering the wavelength dependence of the respective retrievals, as specified in the corresponding aerosol models (Torres et al., 2005). In the present study, we have employed the monthly mean, Level-3 AAODs (500 nm) as the background data (for constructing merged AAOD product), which hereafter referred to as SR AAOD.

### 2.3 AOD from ground-based sun photometer network (GR AOD)

Ground-based measurements of AOD used in this study are obtained from the ARFINET observatories (Moorthy et al., 2013; Babu et al., 2013) established by Indian Space Research Organization (ISRO) as well as from the AERONET observatories established (over Indian region) and operated by different institutions jointly with NASA (Holben et al., 1998). The locations of these 44 observatories, the AOD values from which are used in this study, are shown in the top panel of figure 1.

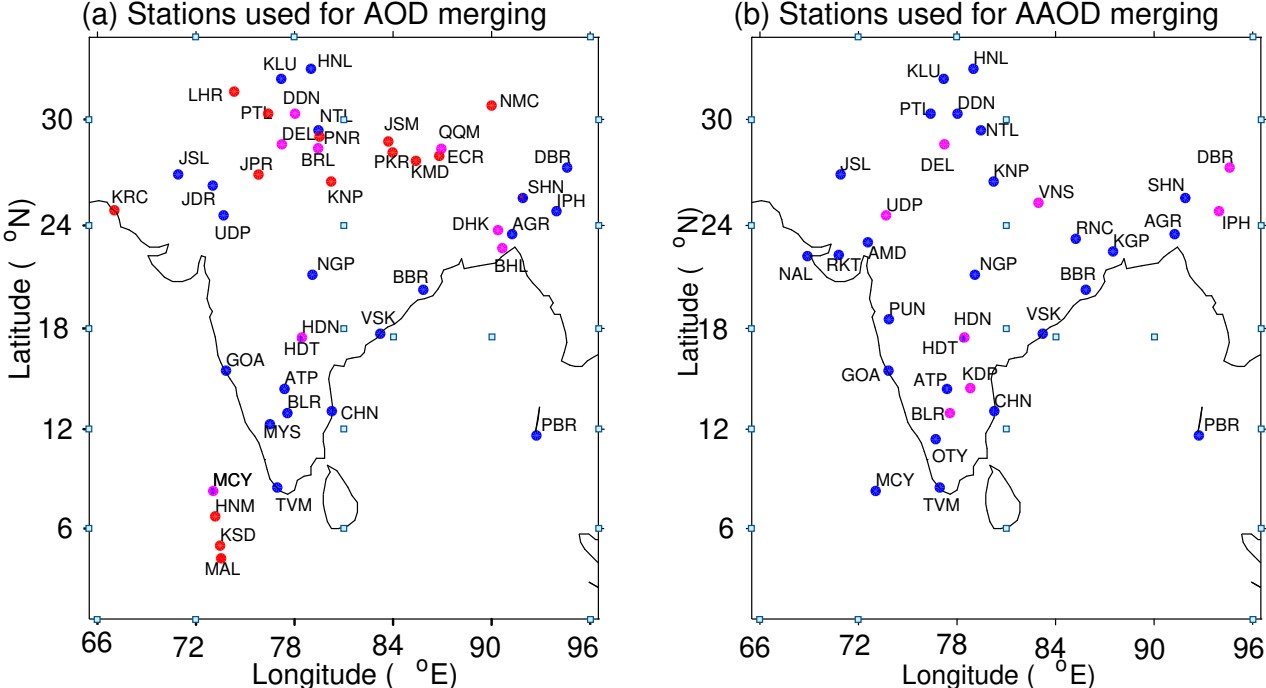

**Figure 1.** Locations of the ground-based stations, AODs (a) and BC (surface level) mass concentration measurements (b) from which are used in the present study. In figure (a), blue and red dots represent respectively the ARFINET and the AERONET stations, the AODs from which are assimilated, while the pink dots represent the stations, the data from which are used for independent validation (and not used in the assimilation). In figure (b), blue and pink dots denote the ARFINET stations providing BC data for assimilation and validation respectively.

The ARFINET observatories are set-up as a part of the project, Aerosol Radiative Forcing over India (ARFI), of ISRO for investigating the spatial-temporal heterogeneities of aerosols, their spectral characteristics, size distributions as well as to assess impact of aerosols on regional radiative forcing. In order to obtain columnar AODs, each observatory in the ARFINET is equipped with either 10 channel Multi Wavelength Radiometer (MWR) (Moorthy et al., 1989, 2013) and/or Microtops Sun photometer (MSP) (Morys et al., 2001). The details of analysis of these data and inter-comparison with other commercial and research level sun-photometers are available in the literature (Shaw et al., 1973; Moorthy et al., 2007b, 2013; Kompalli et al., 2010). The present study utilizes the monthly mean AOD data at 500 nm, measured at ARFINET stations which are detailed in Table S1 provided in supplementary material. The AERONET stations (Table S2 from supplementary material) comprised of a network of automatic, sun-sky scanning radiometers; is set up and supervised by NASA (Holben et al., 1998, 2001). The present study employs level-2, monthly mean AODs at 500 nm provided by AERONET (*https://aeronet.gsfc.nasa.gov/*).

## 2.4 AAOD from ground-based BC measurements

Unlike AOD, there are no direct ground-based measurements for absorption AOD. Hence, in order to construct a reliable dataset of AAOD over Indian region, we have employed the regular BC mass concentration measurements (at surface level) performed

at 34 ARFINET observatories (figure 1b). The list of these observatories along with their geographical co-ordinates and broad geographical features of the respective regions, is provided in the Table S3 (from supplementary material). The reason behind using the BC measurements for the current purpose lies in the fact that black carbon is the primary light absorbing aerosol specie not only because of its ability to absorb the radiations over a wide wavelength range but also due to its longer atmospheric residence time (of the order of few days to weeks) in the lower troposphere (Babu and Moorthy, 2002). In addition, BC can alter the properties of other aerosol species by mixing with them (Jacobson, 2001). The other strong absorbing aerosol specie over this region is the mineral dust, which is perennially present, especially over the north-western arid regions and Indo-Gangetic Plains (IGP).

The continuous measurements for BC mass concentration are performed using the Aethalometer (from Magee Scientific Inc., USA) (Hansen et al., 1984; Hansen and Novakov, 1990) at these 34 ARFINET stations (figure 1b). For maintaining consistency in measurements and ensuring data quality, these Aethalometers are operated under a common protocol and are periodically inter-compared.

The measured BC mass concentrations are used to estimate the AOD due to BC, making use of the Optical Properties of Aerosols and Clouds (OPAC) model (Hess et al., 1998). Presently, there is no empirical model available for the vertical distribution of BC over the Indian region. As such, in order to specify the vertical distribution of BC in OPAC, we have considered the representative vertical distribution of BC, based on commonly observed features of vertical heterogeneities of aerosols reported over the Indian region (Satheesh et al., 1999, 2008; Babu et al., 2011). Due to vertical mixing caused by the eddies within the day-time convective boundary layer, aerosols can be considered to be near uniformly distributed within the Planetary Boundary Layer (PBL). Aircraft and balloon measurements of BC over different regions of India and during different seasons (Satheesh et al., 2008; Suresh Babu et al., 2010; Babu et al., 2011) have shown that during daytime the near uniform BC mass concentrations are observed till $\approx 2$ km. Accordingly, we have considered the uniform mass concentration of BC within the PBL and an exponential decaying above it following the scale heights (seasonally varying) reported by Yu et al. (2010) using CALIPSO observations over the Indian region.

For deriving the BC AODs from BC mass concentration using OPAC, the PBL heights (PBLH) provided by the Modern-Era Retrospective analysis for Research and Applications-2 (MERRA2) reanalysis dataset (Gelaro et al., 2017), are used. The MERRA2 PBLH dataset has been validated by comparing with PBLH derived from radiosonde and GPS radio occultation measurements over Indian region by Sathyanadh et al. (2017). Depending on the location, the slope of a linear fit between the MERRA2 derived and measured PBLH is varying between 0.75 to 0.93 (Sathyanadh et al., 2017). Nevertheless, the validation exercise performed by (Sathyanadh et al., 2017) is representative of a quite limited period (May to September 2011). Therefore, we have validated MERRA-2 PBLH with those estimated from radiosonde measurements (downloaded from *http://weather.uwyo.edu/upperair/sounding.html*) performed at eight representative locations (figure 2), during the period of year 2008 to 2018.

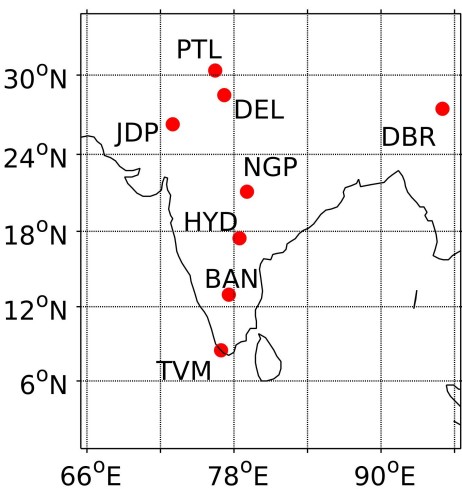

**Figure 2.** Locations of the ground stations, radiosonde measurements from which are used for the purpose of validating PBLH derived by MERRA-2. These subregional representative stations form a subset of ground-based observatories, AOD and BC mass concentration measurements from which are employed for construction of assimilated AOD and Absorption AOD (AAOD) products.

The scatter plots between spatially collocated MERRA-2 PBLH and those derived from radiosonde measurements over the eight locations, are presented in figure 3. It can be seen from figure 3 that, MERRA-2 PBLH are well-correlated with those estimated from radiosonde measurements, although the correlation coefficient varies from 0.63 to 0.96, w.r.t the location. The details about this validation exercise are provided in Appendix C.

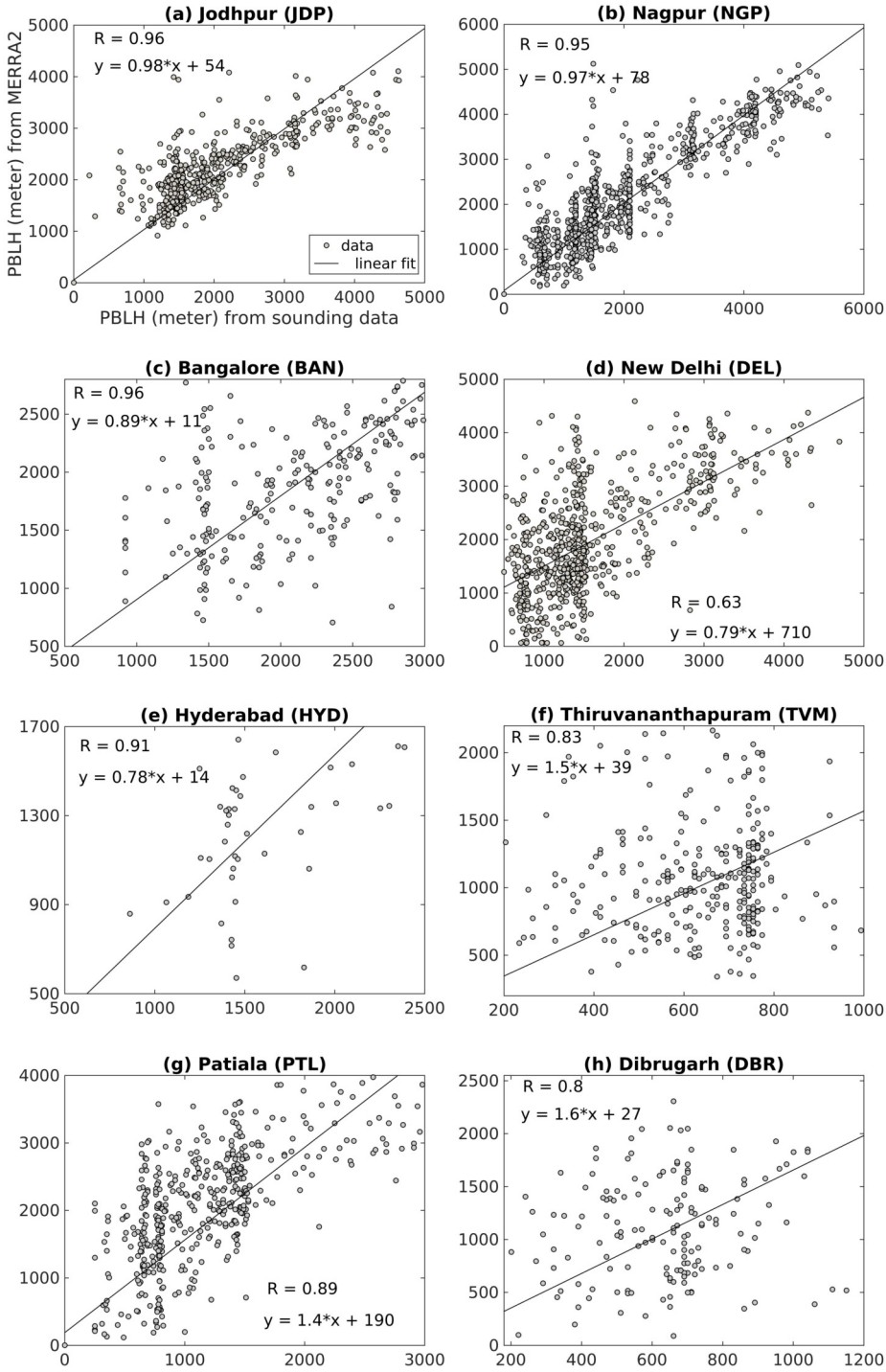

**Figure 3.** Comparison of spatio-temporally collocated MERRA-2 PBLH with those derived from radiosonde measurements performed at 8 representative locations during year 2008 to 2018. The correlation coefficient (R) (significant at 95% confidence limit) and the equation of linear regression between the two PBLH estimates are provided in each of the figures.

After deriving BC AODs by incorporating BC mass concentration measurements and MERRA-2 PBLH in OPAC (Hess et al., 1998), the corresponding absorption AODs contributed by BC are then estimated by considering BC SSA as 0.22 (Hess et al., 1998).

Mineral dust aerosols form another important species contributing to aerosol absorption, not only the solar but also the outgoing terrestrial radiation (Satheesh et al., 2007; Deepshikha et al., 2006a, b). Mineral dust prevails over the central and northern parts of Indian region; locally produced as well as advected from west Asian and east African regions (Moorthy et al., 2005; Niranjan et al., 2007; Beegum et al., 2008). As a first step to estimate the dust absorption optical depth, we have computed the Infrared Difference Dust Index (IDDI) (Legrand et al., 2001), the reduction in infrared radiance sensed by the satellite-based instruments, due to atmospheric dust aerosols. Mathematically, IDDI is defined as shown in equation 1 following Legrand et al. (2001).

$$IDDI = RD \uparrow - RC \uparrow \tag{1}$$

Here, $RD \uparrow$ and $RC \uparrow$ respectively denote the outgoing long-wave terrestrial radiations in dust-loaded and clear sky (i.e. no aerosol and no clouds) conditions at top of atmosphere (TOA). Thus, IDDI is an indicator of amount of columnar dust loading in the atmosphere (Tanré and Legrand, 1991; Legrand et al., 2001). In this study, we estimated the IDDI from the brightness temperature corresponding to IR radiance (10.5 to 12.5 microns) measured by the Very High Resolution Radiometer (VHRR) aboard geostationary Indian satellites, Kalpana-1 and INSAT-3A following Deepshikha et al. (2006a, b) and Srivastava et al. (2011). The daily estimates of IDDI are then used to infer the dust AOD (500 nm) (following Srivastava et al. (2011)) from which dust absorption AOD are estimated by considering dust SSA (at 500 nm) as 0.91 (Moorthy et al., 2007a). The monthly mean dust AAODs are then interpolated to the locations of ARFINET laboratories from which the BC measurements are employed. The final AAODs to be merged with the OMI AAODs are then constructed as shown in equation 2 and are henceforth referred as GR AAOD.

$$GR\ AAOD = BC\ AAOD + dust\ AAOD \tag{2}$$

## 3 Merging the different datasets: Assimilation Methodology

In generating merged datasets for AOD and AAOD by assimilating different datasets, we have adopted widely accepted statistical data assimilation techniques, along with a few physical constraints; as detailed below.

Several methods are available in the literature for combining scattered observations with gridded data, including variance minimization based methods (like 3D-VAR) and heuristic methods like Successive Correction Methods (SCM) (Kalnay, 2003; Lewis et al., 2006). Methods based on variance minimization are mathematically sophisticated and perform the assimilation in such a way that the variance of entire assimilated field is guaranteed to be lesser than those of the parent datasets. On the other hand, SCM are empirical in nature and blend in the observations with background data locally within a specified radius of influence, although it does not assure the variance minimization.

As such, Mitra et al. (2003, 2009) have employed Cressman method (Cressman, 1959), a variant of SCM, for forming daily analyses of rainfall over Indian region by merging rain gauge measurements with satellite retrieved precipitation. For AOD assimilation, the similar method has been employed by Chung et al. (2005), while Optimal Interpolation (OI) has been used by Collins et al. (2001); Adhikary et al. (2008) and Singh et al. (2017). Further, 3D-VAR has been employed by Niu et al. (2008) and Zhang et al. (2008) for assimilation of dust aerosol properties and AODs respectively. Nevertheless, analysed AODs produced by either variance minimization methods or SCM based methods, need not be always bounded by their parents.

In the present study, the datasets to be merged represent the same parameter although measured by different techniques. The ground based measurements comprise of data from dense network of observatories representing all distinct environments over the region and has sufficient temporal coverage to smooth out any isolated events or episodes. Ground-based measurements also provide a fairly accurate spatio-temporal distribution; while the satellite products provide wide spatial coverage. In view of this, it is logical that merged datasets be bounded by the respective parent datasets. This also leads to a fairly smooth spatial variation that are needed for inputting the merged products to climate impact assessment models, without compromising on the accuracy of distinct spatial features.

### 3.1 Merging methodology for AOD

In order to achieve the variance minimisation while ensuring the merged AODs to be bounded by the parent datasets (i.e. SR and GR AOD), we have employed SCM, with a variation, which we refer as Weighted Interpolation Method (WIM). It expresses the merged AODs as weighted average of SR and GR AODs, in such a way that the resulting merged AOD values are always bounded by the parents. While performing the weighted average, the weight given to GR AOD is inversely proportional to the distance between location corresponding to a given grid point and a ground-based observatory, following the Cressman (1959). This inverse distance weighting method enables the merged AOD at a given location to be largely represented by the ground-based measurements from nearer stations vis-a-vis farther ones. Depending on the weights given to GR AOD, WIM assigns weights for SR AOD such that the sum of weights for SR and GR AODs is always unity. This ensures that merged AODs are bounded by ground-based and satellite retrieved AODs. The weighted average of SR and GR AODs is then performed in an iterative manner, till the merged AODs interpolated at the locations of ground-based observatories matches with respective GR AODs within its uncertainty limits. Mathematically, WIM is expressed as shown in equation 3.

$$X_{k+1} = R[X_k] + QW[Z] \tag{3}$$

Here, $X_{k+1}$ = vector (size $n \times 1$) of merged data at $k + 1^{\text{th}}$ iteration, where $n$ represents total number of grid points in the domain. Mathematically, $n = n_x \times n_y$ where $n_x$ and $n_y$ denote number of nodes in longitudes and latitudes respectively. Further, $X_k$ represents vector (size $n \times 1$) of merged data at $k^{th}$ iteration. During first iteration of equation 3, $X_k$ is equal to vector of background data as provided by SR AOD. $Z$ refers to vector (size $m \times 1$) of GR AOD; where $m$ represents number of ground-based measurements available at that instant in the whole spatial domain. $H$ is an interpolation matrix (size $m \times n$) which bi-linearly interpolates the gridded satellite data to the locations of ground-based observatories. The details about construction of $H$ can be found in (Kalnay, 2003; Lewis et al., 2006).

As the satellite retrieved and ground-based AODs are not collocated, one needs to give appropriate weights to SR and GR AOD values during merging them. In equation 3, $QW$ (size $n \times m$) is the normalized weight matrix for GR AODs, and R (size $n \times n$) is the weight matrix for SR AODs. The normalised weight matrix is constructed as a product of two matrices; $Q$, the normalisation matrix (size $n \times n$) and $W$, the weight matrix (size $n \times m$) for GR AODs. The weights given to GR AODs, which form elements of matrix $W$ are computed using one of the widely-accepted inverse-distance method which is given by Cressman (Cressman, 1959) as shown in equation 4.

$$W_{ij} = \begin{cases} \dfrac{d^2 - r_{ij}^2}{d^2 + r_{ij}^2}, & \text{if } r_{ij} \leq d \\ 0, & \text{otherwise} \end{cases} \tag{4}$$

Here, $W_{ij}$ denotes the weight given to GR AOD from $j^{th}$ ground-based observation location during merging it with SR AOD at $i^{th}$ grid point. This weighting strategy (equation 4) ensures that the contribution of ground-based measurements to merged AOD is higher (lesser) if the distance between a ground station and a grid point (referred as $r_{ij}$ in equation 4) is lesser (higher). The weight matrix $W$ also makes sure that merged AODs are not contributed by the GR AODs from the ground-based stations lying outside the radius of influence which is denoted by $d$ in equation 4. It is to be noted here that the radius of influence corresponds to the region surrounding a given location, within which an AOD from that location can be considered to be largely representative. The details about the choice of radius of influence for the present study are described in section 3.0.2.

The weight matrix $(W)$ thus computed (equation 4) needs to be normalized to make sure that sum of the weights given to GR AODs from all the stations, is less than unity. This normalising process thus constrains the merged AODs by the available ground-based measurements. In order to perform this normalization, the diagonal matrix, $Q$ (which multiplies to $W$ as shown in equation 3) is constructed as follows (Cressman, 1959; Kalnay, 2003; Lewis et al., 2006).

$$Q_{ii}^{-1} = \left[ \sum_{j=1}^{m} W_{ij} + \frac{\sigma_o^2}{\sigma_B^2} \right] \tag{5}$$

The first term on the rhs of equation 5 is the summation over the weights given to all GR AODs from all the stations within radius of influence from $i^{th}$ grid point and 2nd term is the ratio of error variances in GR AOD ($\sigma_o^2$) to that in SR AODs ($\sigma_B^2$). This normalization strategy also ensures that the weights for a parent dataset reduce with the increase in its uncertainty. As major portion of background data is formed by MODIS AODs, $\sigma_B$ is represented as rms (root mean square) uncertainty in MODIS AODs, which is given as $0.03 + 0.2\tau_{sat}$ by Sayer et al. (2013), where $\tau_{sat}$ is the MODIS AOD. The $\sigma_o$ term is formed by uncertainty of GR AOD measurements made at ARFINET and AERONET observatories. As the uncertainties in GR AODs at different wavelengths are in the range of 0.01 to 0.03 (Holben et al., 1998; Babu et al., 2013), the maximum uncertainty (i.e.0.03) is considered as $\sigma_o$ .

After computing the normalised weights for GR AOD, we have calculated the weights to the SR AODs such that the sum of the weights for SR and GR AODs is unity. In other words, the weights for SR AODs are computed such that the merged AODs are guaranteed to be a convex combination of the parent datasets. Mathematically

$$R_{ii} = I_{ii} - \sum_{j=1}^{m} [Q_{ii} W_{ij}] \tag{6}$$

Here, $R$ is the diagonal matrix with its $i^{th}$ diagonal element referring to the weight given to SR AOD at grid point at $i^{th}$ grid point. $I$ is the Identity matrix of size $n \times n$.

It is also to be mentioned here that WIM (equation 3) makes sure that uncertainties in MG AODs are either less than uncertainties in SR and GR AODs or at least less than the largest of the two. The theoretical proof for this is given in Appendix A.

### 3.1.1 Modified Weight Matrix formulation

The weight matrix formulation (equation 2) involves the distance ($r_{ij}$) between a grid point and a network observatory as well as the radius of influence ($d$) from that grid point. However, both of these quantities ($r_{ij}$ and $d$) can be computed/estimated based on either only horizontal or horizontal and vertical coordinates of corresponding grid point and ground station, depending upon the nature of problem. In the current problem of AOD merging, apart from horizontal distance, it is essential to consider the altitude difference between a grid point and a ground station. This is because, AODs measured at an aerosol observatory located over a sharp peak situated over a large plain terrain may not be representative of AOD corresponding to adjoining grid points over the plains due to sharper variations in aerosol concentrations in vertical than horizontal direction. Hence, in order to have a realistic merging, especially in cases involving merging of AODs from two locations within the horizontal radius of influence yet differing in altitudes significantly, it is necessary to take into account the vertical distribution of aerosols and an associated length scale.

Due to the dynamics of the day time convective boundary layer and the associated updrafts, aerosols can be considered to be near uniformly distributed within the planetary boundary layer (PBL); but above this, vertical heterogeneities are possible (for example Satheesh et al. (2008); Suresh Babu et al. (2010); Babu et al. (2011)). As such, the planetary boundary layer height is considered as the region within which the aerosol distribution is near-homogeneous in the vertical. However, the vertical gradients in aerosols could be much sharper than the horizontal variations, especially above the PBL. Concentration of aerosols may significantly differ above the top of the PBL, which acts as a virtual lid (leaky although) shielding the free troposphere from surface based emissions, significantly. A typical example of such case is the Nainital station located over the mountain peak of nearly 2 km elevation above mean sea-level, at a radial distance of $< 50$ km from the Gangetic plains on the south, east and west of it. Similar is the case with a few other stations such as Shillong, Ooty etc..

As such, we expressed the weight matrix ($W$) in equation (3) as the product of two matrices (scalar product) of the same order, $W1$ and $W2$ which take into account the horizontal and vertical variation of aerosols respectively. The details are as given below.

Horizontal component of weight matrix ($W1$) : This weight matrix (size $n \times m$) is defined in terms of horizontal radius of influence ($d$) and the horizontal separation between a grid point and a ground network observatory ($r_{ij}$), as given below in equation 7.

$$W1_{ij} = \begin{cases} \dfrac{d^2 - r_{ij}^2}{d^2 + r_{ij}^2}, & \text{if } r_{ij} \leq d \\ 0, & \text{otherwise} \end{cases} \tag{7}$$

In the present study, the radius of influence is considered to be 250 km (for the first iteration of equation 3) following Winker
et al. (1996) who have suggested the global horizontal correlation length scale for aerosol to be  200 km, using observations from The Lidar In-space Technology Experiment (LITE). In the current work, the radius of influence is reduced by 50 km during each successive iteration of equation 3, to make sure that GR AODs from the location nearest to the given grid point is merged with the background data to the maximum possible extent while iterations converge.

The vertical component of the weight matrix ($W2$) (size $n \times m$) is configured in terms of height of influence ($H$) and the
altitude difference between a grid point and a network observatory ($h_{ij}$) as given below.

$$W2_{ij} = \begin{cases} \dfrac{H^2 - h_{ij}^2}{H^2 + h_{ij}^2}, & \text{if } h_{ij} \leq H \\ 0, & \text{otherwise} \end{cases} \tag{8}$$

Here, $h_{ij}$ represents the difference between altitudes of $i^{th}$ grid point and $j^{th}$ observation location. $H$ is the height of influence defined as PBLH $+\tau$ , where $\tau$ is the height of layer measured above PBL and in which the aerosol concentrations are considered to be decreasing rapidly from the near constant value within PBLH which is specified using MERRA2 reanalysis dataset. The value of $\tau$ has been taken from the variance of PBLH given by MERRA2. For this, we computed the covariance
matrix (size $n \times n$) from monthly mean PBLH over the Indian region. The diagonal elements of this matrix provide variances in the PBLH data at each grid point. After computing standard deviations ($\sigma$) from variance values, $\tau$ values are taken as $2\sigma$. Based on relation between $H$, $h_{ij}$ and PBLH, any of the three following cases can arise, with each defining the distinctive way in which $W1$ and $W2$ contribute to the resultant weight matrix (i.e. $W$ ).

1. **If $h_{ij} \leq$ PBLH**
In this case, $j^{th}$ ground station is located within the PBL of the $i^{th}$ grid. In accordance with the consideration of well mixed boundary layer, $W2$ would have the highest weight (=1) and the resultant weight matrix would be solely determined by its horizontal component (equation 7). This leads to an element of the resultant weight matrix being expressed as shown in equation 9.

$$W_{ij} = W1_{ij} \tag{9}$$

2. **If $H \geq h_{ij} >$ PBLH**
In this case, $j^{th}$ ground station is at an altitude just above the PBL of the $i^{th}$ grid, but not high enough to be considered

uninfluenced by variations at the $i^{th}$ grid point, rather its influence would be rapidly decreasing. In this case, the resultant weight matrix $W$ is expressed as the scalar product of $W1$ and $W2$, so that

$$W_{ij} = W1_{ij} * W2_{ij} \tag{10}$$

3. **If $h_{ij} >$ PBLH**

   In this case, altitude of $j^{th}$ ground station is high enough above the PBL at the $i^{th}$ grid point such that AOD measured at such ground station has hardly any relevance to the AOD at $i^{th}$ grid point. As such, the grid point and ground station are considered to be independent and $W2$ is set to zero and hence the resultant weight matrix, which is the product of $W1$ and $W2$ becomes zero (equation 11).

$$W_{ij} = 0 \tag{11}$$

After constructing $W$ following the above considerations, normalizing matrix $Q$ is computed by substituting corresponding $W$ into equation 5. This is followed by computation of weight matrix for background data, R as given in equation 6. Following the construction of $W$, $Q$ and $R$ matrices, equation 3 is solved iteratively with SR AOD being the background data for the first iteration. The solution of the first iteration forms the background data for the second iteration and the procedure is repeated until the norm of vector of absolute difference between GR AOD and merged AOD (MG AOD) interpolated to locations of ground observatories (mathematically, norm($Z - HX_k$)), reaches a pre-set limiting value of 0.02, which is the mean uncertainty in ground-measured AODs (Holben et al., 1998; Babu et al., 2013). In the present study, this condition is satisfied within $\approx 10$ iterations. Nevertheless, in some of the cases, norm($Z - HX_k$) gets levelled off before reducing to the limiting error value. This occurs due to AOD measurements at some of the ground stations being unassociated with AODs corresponding to grid points surrounding them. In such cases, iterations of equation 3 are performed till absolute difference between errors (i.e. norm($Z - HX_k$) during successive iterations is less than $10^{-3}$.

### 3.2 Merging datasets for AAODs

For AAOD merging, the method slightly differed from the above, as the ground based observations are only of the BC mass concentration measurements which are representative only of surface level black carbon, unlike AOD which has been columnar for both space-based and ground-based measurements. As detailed in section 2.4, the columnar absorption optical depth for BC are estimated by incorporating the BC number concentration (corresponding to BC mass concentration measurements) into OPAC (Hess et al., 1998) in which the vertical distribution of BC is specified using commonly observed characteristics of vertical heterogeneities of aerosols reported over the Indian region (Satheesh et al., 1999, 2008; Babu et al., 2011). In addition to BC, the contribution of dust is also taken into account to construct the columnar AAODs to be merged with OMI AAODs (section 2.4).

Unlike the ground-based AODs demonstrating much stronger correlation (R = 0.77, figure 4a) with satellite AODs, the above mentioned GR AAODs are relatively weakly correlated with OMI AAODs (R = 0.35, figure 4c). Owing to these differences,

employing WIM for AAOD assimilation is observed to generate non-smooth and highly discontinuous merging patterns. As such, we have employed one of the widely used data assimilations methods, 3D-VAR (Niu et al., 2008; Zhang et al., 2008), which is based on principle of least-squared error minimization. In 3D-VAR, the merged AAODs are estimated as a solution of the minimizer of the following objective function (referred as J) which expresses the weighted sum of the departures in merged AAODs from GR and OMI AAODs, as shown in the equation 12.

$$J(x) = \frac{1}{2} \left[ (X - X_b)^T B^{-1} (X - X_b) + (Z - HX)^T O^{-1} (Z - HX) \right] \tag{12}$$

Here, $X$ and $X_b$ refer to MG AAOD and OMI AAOD vectors, respectively, of size $n \times 1$, where $n$ is the total no of grid points in the spatial domain (n=1120 for the current study). Further in equation 12, $Z$ denotes the vector of GR AAODs, which is of size $m \times 1$ where $m$ is the no. of ground based observatories, data from which are available during the respective month. The map between the grid space and the observation space is provided by the interpolation matrix referred as $H$ (size $m \times n$) (equation 12) (Lewis et al., 2006; Kalnay, 2003). Finally, $B$ (size $n \times n$) and $O$ (size $m \times m$) represent the error covariance matrices for OMI AAODs and GR AAODs respectively. The minimizer to the above mentioned objective function is estimated by solving the following equation 13.

$$\left[ B^{-1} + H^T R^{-1} H \right] X = \left[ B^{-1} X_b + H^T R^{-1} Z \right] \tag{13}$$

The further details about the 3D-VAR can be found in Kalnay (2003); Lewis et al. (2006).

Constructing error covariance matrices ($B$ and $O$) is a fundamental element of 3D-VAR data assimilation. This is mainly because, the underlying correlation structure and the actual variance values not only dictate the pattern in which observations get merged with the background data but also decides the weights given to each of the parent datasets during the merging process. In the present study, the observation error covariance matrix ($O$) is considered to be diagonal implying that errors in GR AAODs from different ground-based stations are uncorrelated, which is generally true and is followed earlier also (Niu et al., 2008; Zhang et al., 2008; Singh et al., 2017). As the diagonal terms of the covariance matrix refer to variance of the corresponding data, the diagonal terms of $O$ are formed by the taking square of uncertainties in the GR AAODs which are estimated as explained below.

It can be understood that, the uncertainties in BC AAOD arise largely from the uncertainties in BC mass concentration measurements, the assumed vertical distribution of BC as well as uncertainties associated with OPAC model. So, in order to estimate the uncertainties in BC AAODs, we perturbed BC mass concentration measurements, MERRA-2 PBLH and scale height within their respective uncertainty limits to compute the multiple realizations for a set of BC AAODs. For this exercise, the uncertainties in BC measurements are considered to be 2 to 5% (Hansen and Novakov, 1990; Babu et al., 2004; Dumka et al., 2010) and those in MERRA2 PBLH are estimated to be 5 to 20% while the uncertainties in scale height for vertical distribution of aerosols (derived from CALIPSO measurements) is considered to be $\approx$100 m, (Kim et al., 2008). The standard deviation of the multiple realizations for a given BC AAOD, is adopted as the uncertainty in the corresponding BC AAOD. This analysis showed that the uncertainties in BC AAODs vary from around 11% to 20% with its mean, i.e. 15% being considered

as the uncertainty in BC AAOD. Similarly, the uncertainties in dust AAOD, which are largely emanating from the uncertainties in vertical heterogeneity of dust and its optical properties are estimated to be around 25% of dust AAOD. The diagonal terms of error covariance matrix for observations ($O$) are thus constructed as shown in following equation 14.

$$O_{ii} = (0.15 * \text{BC AAOD}_i)^2 + (0.25 * \text{dust AAOD}_i)^2 \tag{14}$$

Here, i represents the index varying from 1 to no. of ground-based stations.

As the observation error covariance matrix is diagonal, the patterns of merging GR AAODs with OMI AAODs are fully dependent on the background error covariance matrix ($B$). In the view of this, we have estimated the background error covariance matrix from historical time-series (2005 to 2016) of OMI AAOD at 500 nm. This covariance matrix provides not only the spatial structure of correlation between OMI AAODs at $n$ grid points as well as the variances of the background data. The details about the construction of seasonally varying $B$ from time-series of OMI AAOD are provided in the section S2 from supplementary material.

It is to be noted that 3D-VAR assures the variance of merged estimates to be lesser than those of both the parent datasets (Lewis et al., 2006; Kalnay, 2003). However, the theoretical proof for variance in analysed (assimilated) estimates (constructed by 3D-VAR) being smaller than those in parent datasets, is provided in Appendix B.

This translates to the uncertainties in the merged AAODs being guaranteed to be smaller than those in OMI AAODs and GR AAODs.

## 4  Results and discussion

Following the above methods, we constructed spatially and temporally homogeneous gridded data products of AOD and AAOD over the study domain, for each month of the year. Before examining the products for their basic features, it is essential to validate them with independent measurements.

### 4.1  Validation of merged products

For the validation purpose, we have evaluated the performance of the merged datasets against independent ground-based measurements from sub-regional representative locations, the data from which did not enter the assimilation process. The merged AOD (AAOD) product constructed by assimilating long-term, ground based AODs (AAODs) from the network of 36 (26) observatories are validated against the AODs (AAODs) from 8 (8) independent, representative observatories, which are shown by pink dots in figure 1. As the grid nodes and locations of ground-based observatories are not collocated, we have interpolated the merged AODs and AAODs from the grid nodes contained by $3° \times 3°$ box surrounding the locations of respective ground locations used for validation. The comparisons of collocated merged AODs and AAODs with the respective, independent ground-based estimates are shown by scatter plots in figure 4b and 5b. To assess the quality improvement due to present assimilation, we have shown the scatter plots of the satellite retrieved AOD and AAOD (interpolated to ground station locations) against those from the corresponding independent measurements, in figure 4a and 5a respectively.

The regression lines and the ideal 1:1 lines are also drawn in the respective panels and the corresponding statistics (regression coefficients and correlation coefficient) are also provided in figure 4 and 5.

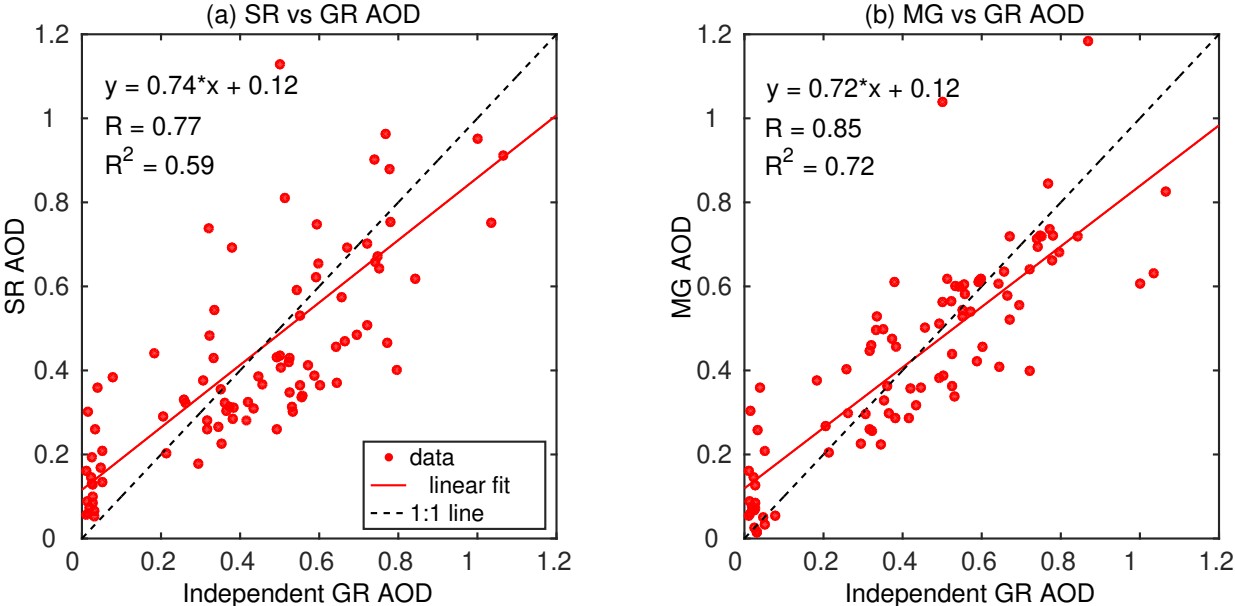

**Figure 4.** Validation of merged AOD product; (a) Satellite AOD vs GR AOD, (b) Merged AOD vs GR AOD. Red lines are regression fitted to the point while the dotted black lines represent the ideal 1:1 case.

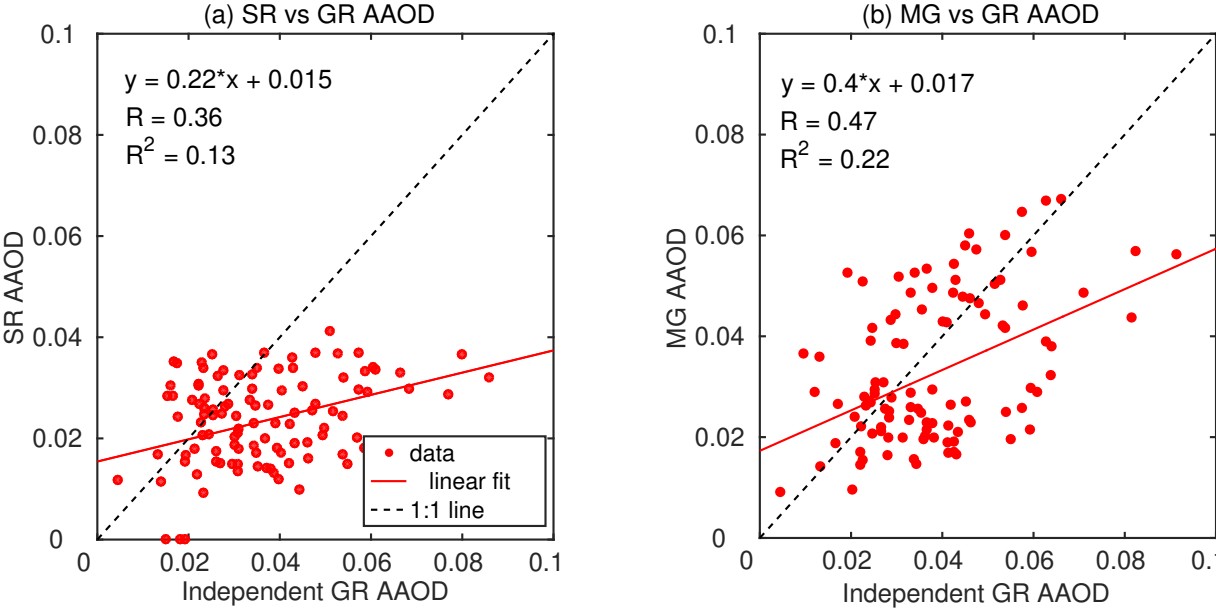

**Figure 5.** Validation of merged AAOD product; (a) Satellite AAOD vs GR AAOD, (b) Merged AAOD vs GR AAOD. Red lines are regression fitted to the point while the dotted black lines represent the ideal 1:1 case.

The merged products are demonstrating improved agreement (stronger correlation) with independent ground-based datasets than that shown by respective satellite products (figure 4,5). This highlights the significant advantage of the assimilation and is all the more important for AAOD (figure 5); the most important parameter for the accurate estimation of atmospheric forcing. With this confidence established through statistical means, we proceeded then to include these independent stations also into the group of ground locations used for merging and the whole assimilation process (section 3.1.1) is repeated to generate the final gridded, merged AOD and AAOD datasets. These final merged AODs and AAODs constructed by assimilating ground-based data from 44 stations for AOD and 34 stations for AAOD, are henceforth referred as MG AOD and MG AAOD respectively.

### 4.2 Spatio-temporal characteristics of merged products

Having generated the harmonized gridded datasets of AOD and AAOD, we examined the spatio-temporal features for their fidelity in reproducing the already reported characteristics over this region from several sub-regional studies. In figure 6 and 7, we present the spatial variation of MG AOD respectively for January-2009 (representative of winter, the season with lowest vertical mixing) and May-2009 (representative of pre-monsoon, summer season, when the convective mixing is very strong). The corresponding features for AAOD are shown in figure 8 and 9. In all the four figures, locations of ground stations, data from which is assimilated, are indicated by circles. In the figure 6 and 7 (figure 8 and 9), the panels from left to right indicate SR AOD (SR AAOD), MG AOD (MG AAOD) and dAOD (dAAOD) which is the difference between merged and satellite-retrieved AOD (SR AAOD).

Mathematically,

$$dAOD = MG\ AOD - SR\ AOD \tag{15}$$

$$dAAOD = MG\ AAOD - SR\ AAOD \tag{16}$$

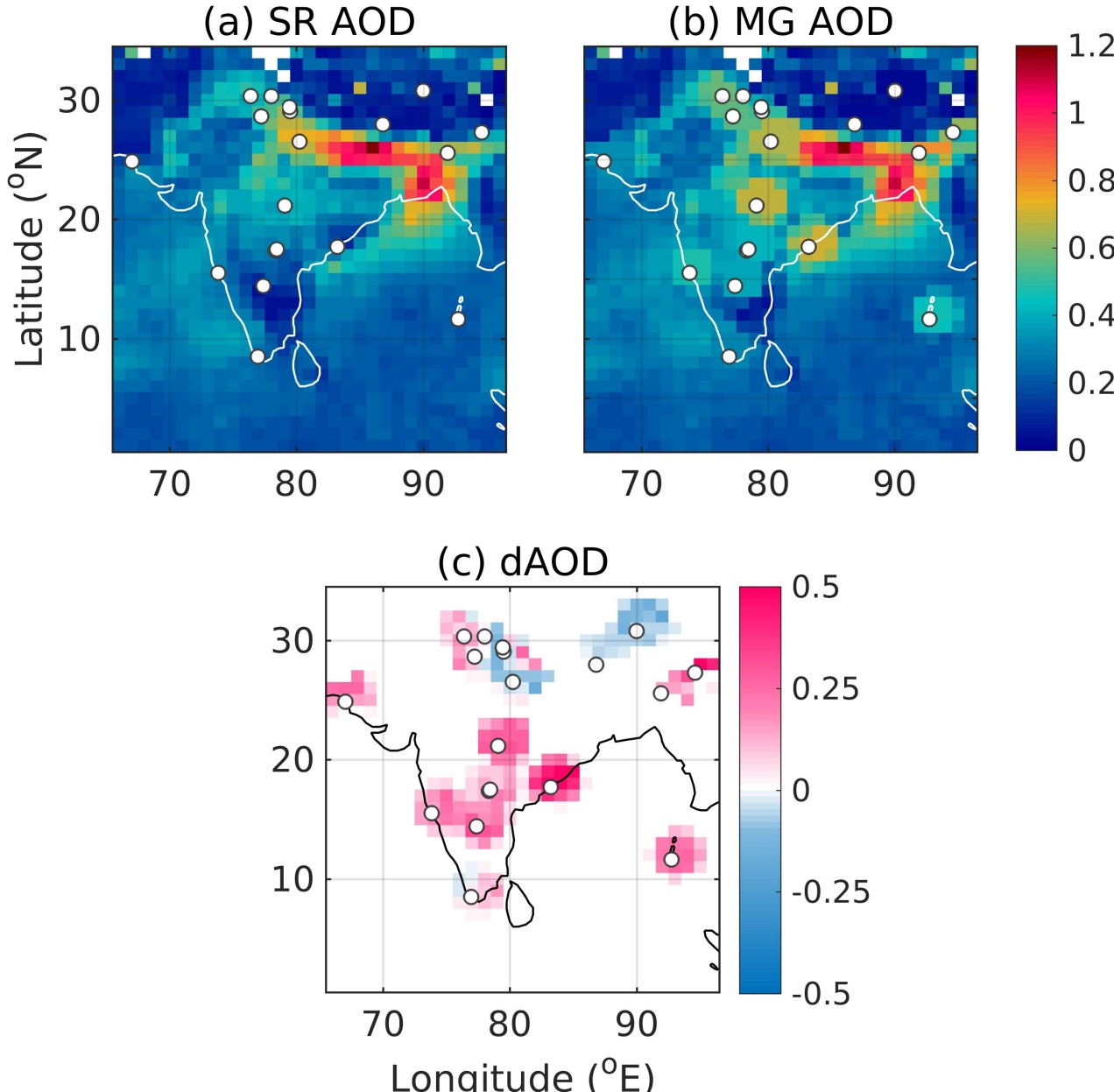

**Figure 6.** Spatial variation of monthly mean SR AOD (a), MG AOD(b) and dAOD (c) for January – 2009.

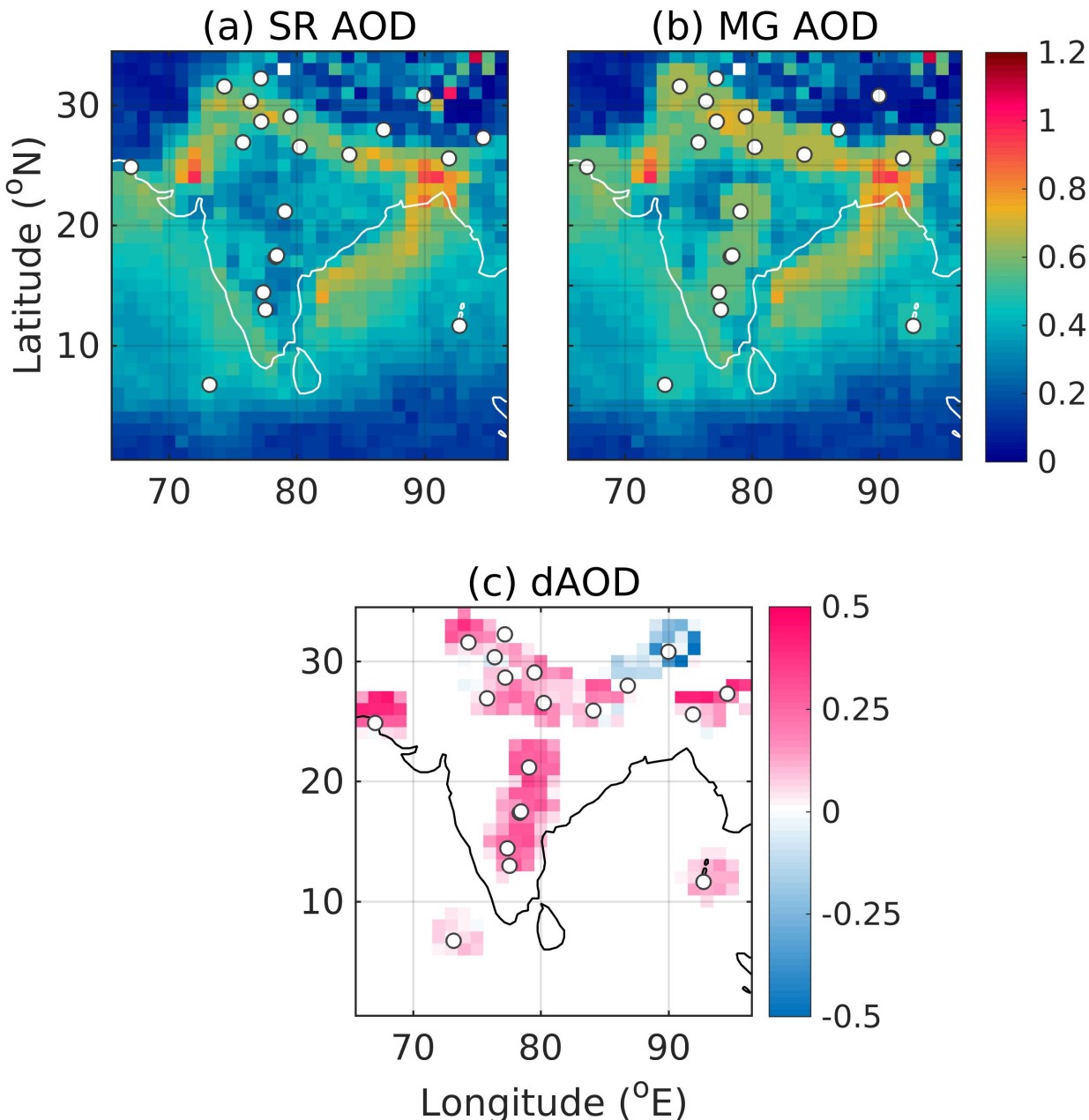

**Figure 7.** Spatial variation of monthly mean SR AOD (a), MG AOD(b) and dAOD (c) for May – 2009.

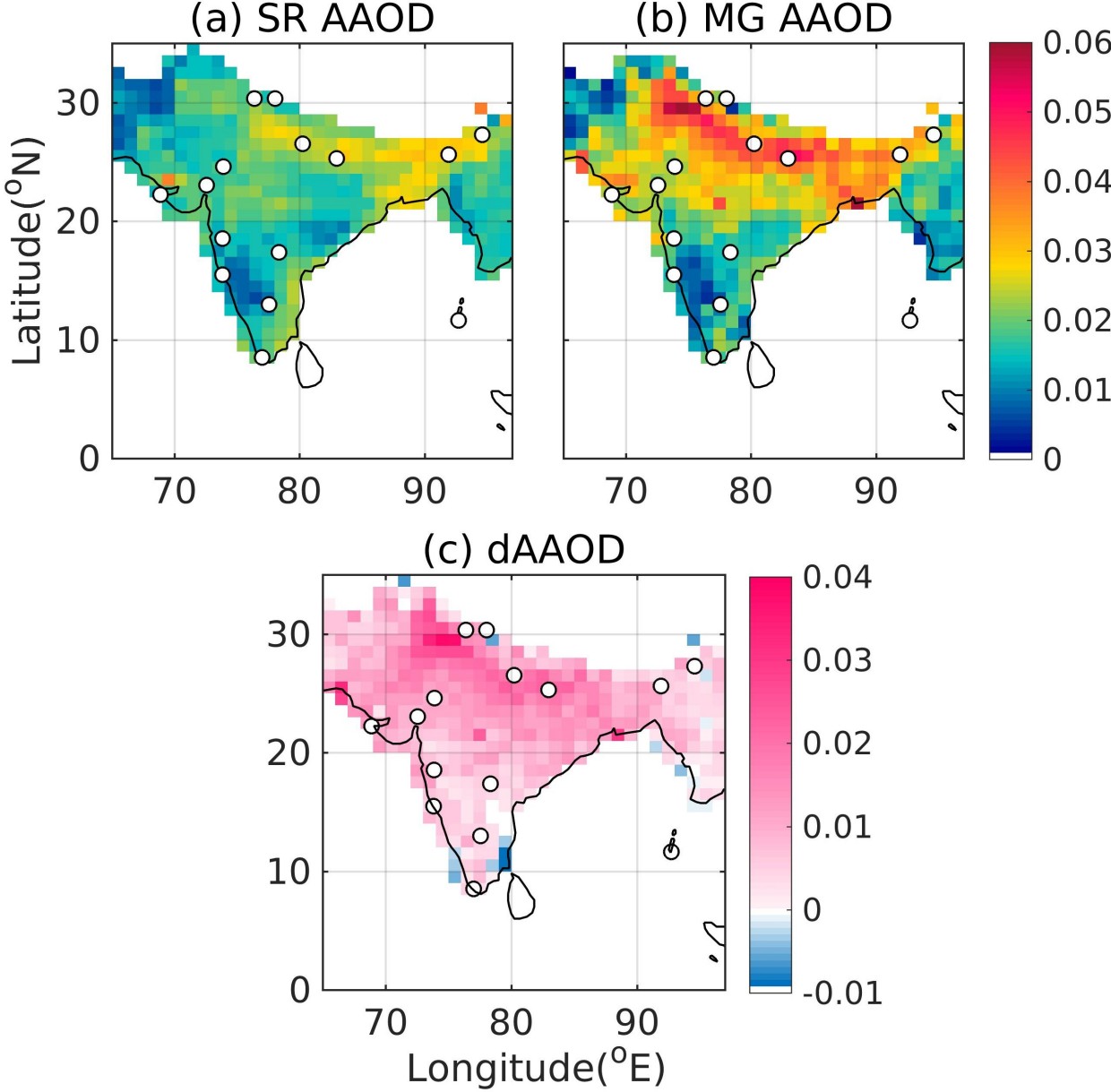

**Figure 8.** Spatial variation of monthly mean SR AAOD (a), MG AAOD(b) and dAAOD (c) for January – 2009.

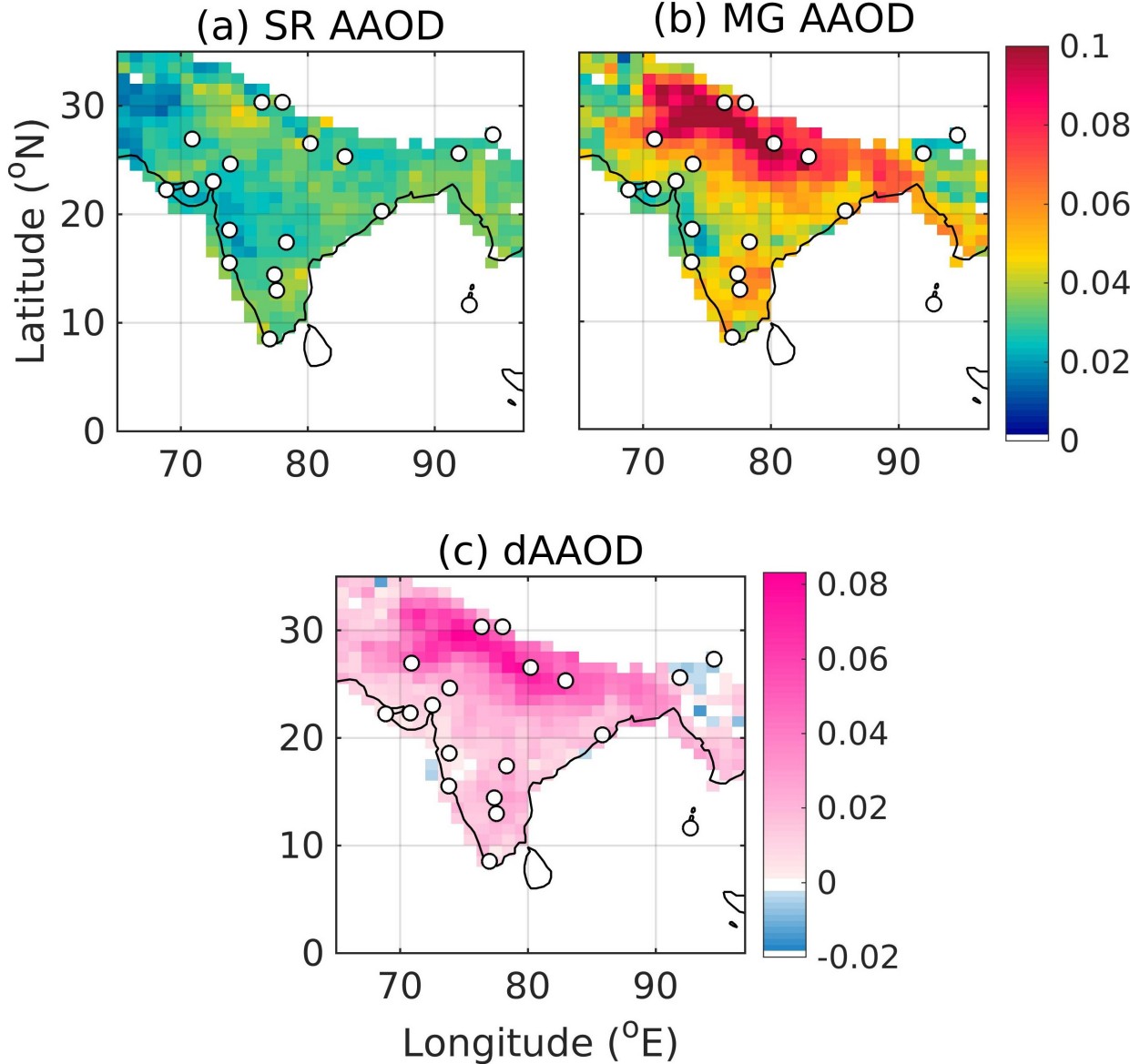

**Figure 9.** Spatial variation of monthly mean SR AAOD (a), MG AAOD(b) and dAAOD (c) for May – 2009.

Figures 6 to 9 clearly show that the broad spatial features are consistent between the merged and respective satellite products. For instance, higher AODs and AAODs are exhibited over Indo-Gangetic plains (IGP) than those over central and peninsular India, by merged products which is in line with their respective gridded parents and also with reports from several past studies (Babu et al., 2013).

5    However, in most of the cases, the merged AODs (Figures 6b and 7b) show higher values than the respective satellite AODs (Figure 6a and 7a) as the ground-station points are approached. This is in line with the general observation about satellite

retrieved AODs being underestimated over this region (Jethva et al., 2005, 2007; Tripathi et al., 2005). However, sufficiently farther away, where no ground-based measurements are available, the merged products tend to be close to the corresponding satellite data. This brings in the need for improving the density of the ground network to further improve the accuracy of regional AOD, for providing even better inputs to climate models.

Further, we estimated the variance in merged AOD and AAODs and compared it with that in respective satellite products. As assured by the assimilation methodologies employed, the uncertainties (square root of variance) in merged AOD and AAODs are observed to be substantially lower than those in the corresponding satellite data. For the above shown representative cases, the uncertainties in merged AODs are observed to be even as small as $\approx 13\%$ of those in SR AOD. The uncertainties in merged AAODs are estimated to be as small as $\approx 82\%$ and $\approx 56\%$ of those in corresponding satellite product, during Jan-2009 and May-2009 respectively.

## 4.3 SSA estimation

The merged, gridded datasets of AOD and AAOD over the domain enable estimation of SSA, the critically important aerosol parameter for radiative forcing estimation. The importance of the accurate estimation of SSA for climate impact assessment of aerosols has been underlined by numerous studies in the past (Haywood and Shine, 1995, 1997; Heintzenberg and Helas, 1997; Russell et al., 2002; Takemura et al., 2002; Loeb and Su, 2010; Babu et al., 2016). Takemura et al. (2002) have shown that the small changes in SSA can even alter the sign of aerosol radiative forcing (ARF) at TOA, while (Loeb and Su, 2010) have demonstrated that uncertainties in SSA could even be the largest contributor to the uncertainties in total direct ARF in clear as well as all sky conditions.

Even though the OMI SSA (Torres et al., 2007) provides a wide spatial coverage, OMI retrievals suffer from uncertainties emanating largely from subpixel cloud contamination as well as assumptions regarding height of an aerosol layer and surface albedo (Satheesh et al., 2009; Jethva et al., 2014; Torres et al., 2007). On the other hand, the highly accurate SSA derived from airborne measurements of scattering and absorption coefficients (Babu et al., 2016), are location and season specific and thus lack the spatio-temporal coverage necessary for the regional ARF estimation. In this context, the merged and validated gridded AOD and AAOD products, generated above, assume importance.

The gridded data for columnar SSA is derived from the merged AODs and AAODs using the relation (equation 17) as follows.

$$\text{SSA} = \frac{\text{MG AOD} - \text{MG AAOD}}{\text{MG AOD}} \tag{17}$$

The spatial variation of the above estimated SSA is presented for the representative months of January-2009 and May-2009, in figure 10.

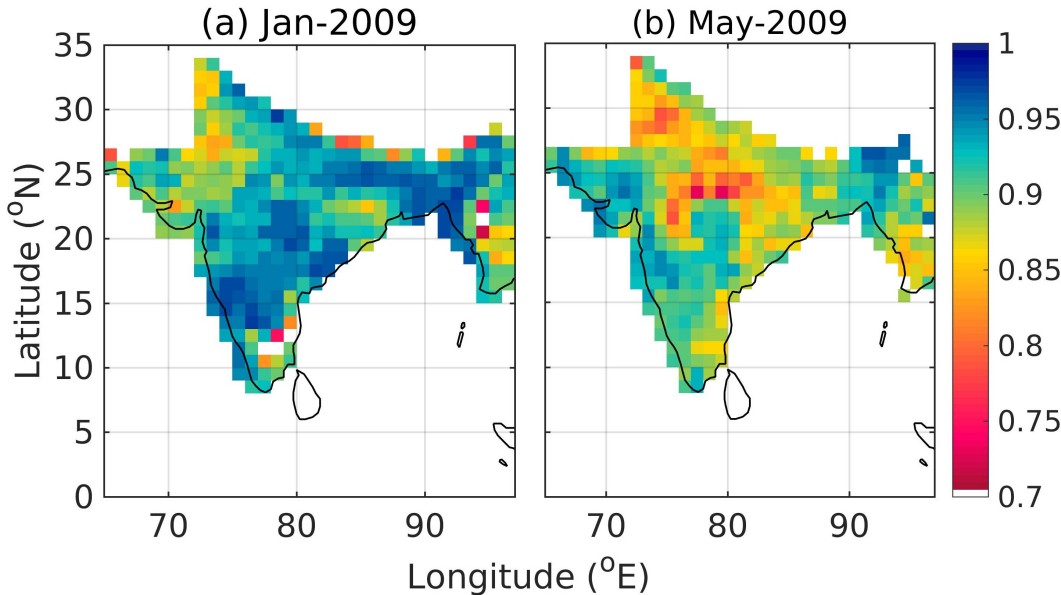

**Figure 10.** Spatial variation of monthly mean columnar SSA estimated using merged AOD and AAODs for January - 2009 (a) and May - 2009 (b).

It can be seen from figure 10 that SSA over Indo-Gangetic plains and northern as well as north-western India is lesser than that over southern India, during both the representative months. This is in line with the regional distribution of SSA reported by Narasimhan and Satheesh (2013) using the gridded SSA retrieved using joint OMI-MODIS algorithm (Satheesh et al., 2009). The lower SSA values over IGP and north-western India indicate higher load of absorbing aerosols, mainly BC (over IGP and northern India) and mineral dust (over north-western Indian region consisting of Thar desert). The increased presence of BC over IGP and northern India can be largely attributed to emission from thermal power plants, increasing number of motorised vehicles as well as biomass burning. Further inspection of figure 10 reveal that eastern coast of India is demonstrating lower SSA vis-a-vis western coast, especially during the pre-monsoonal month of May-2009 (figure 10b). In addition, consistently lower SSA can also be seen over the parts of Myanmar and surrounding regions, during both the representative months (figure 10).

On the background of sensitivity of aerosol radiative effect to the changes in SSA (Haywood and Shine, 1995, 1997; Heintzenberg and Helas, 1997; Russell et al., 2002; Takemura et al., 2002; Loeb and Su, 2010; Babu et al., 2016), it would be imperative to assess the uncertainty in the above estimated SSA (equation 17). For this purpose, we have perturbed MG AOD and AAOD within their respective uncertainty limits to derive multiple realizations for a given SSA (using equation 17) and the standard deviation across these multiple realizations is adopted as an uncertainty in the respective SSA. For the above shown representative cases (figure 10), the RMS uncertainty in SSA is $0.03$ and $0.02$ for Jan-2009 and May-2009 respectively, which is less than that in OMI SSA ($0.05$ to $0.1$)(Torres et al., 2002) and comparable to that in AERONET SSA ($\approx 0.03$ for $AOD_{440nm} > 0.2$ and solar zenith angle larger than $50°$ ) (Dubovik et al., 2000).

### 4.3.1   Seasonality in SSA

In view of the known seasonality in aerosol types arising from the seasonal nature of aerosol sources, transport pathways and the meso-scale and synoptic meteorology, it would be important to examine the seasonality of SSA over the study domain, in the light of already published data. For this, we have considered four representative and fairly homogeneous subregions of the Indian domain to assess the seasonality, as shown in Table 1 (and depicted in figure S1 from supplementary material). In line with the seasonal variation in synoptic meteorology influencing the aerosol field over the Indian region, we have considered three seasons, pre-monsoon which comprises of Mar-Apr-May months and referred as PrM (characterized by strong heating, deeper planetary boundary layer and prevailing westerlies over the region); winter season comprising of Dec-Jan-Feb months (characterized by relatively lesser solar hearing, shallower planetary boundary layers and easterly winds) and post-monsoon (referred as PoM) formed by Oct-Nov months, which mark the transition from summer monsoon to winter season.

**Table 1.** Details of subregions considered

| Sr no. | Subregion ID | Subregion name | Broad geographical characteristics | Latitudinal boundaries in deg. North | Longitudinal boundaries in deg. East |
|--------|--------------|----------------|-----------------------------------|--------------------------------------|--------------------------------------|
| 1 | IGP | Indo-Gangetic plains | Plain plateau | 24.5–28.5 | 78.5–83.5 |
| 2 | NE | North-Eastern India | Mountainous | 23.5–27.5 | 90.5–95.5 |
| 3 | WAR | Western Arid Regions | Arid | 23.5–28.5 | 70.5–76.5 |
| 4 | PI | Peninsular India | Coastal and plain | 7.5–17.5 | 74.5–80.5 |

The SSA values derived from the merged datasets following equation 17, are averaged over the sub-regions (table 1) and seasons are presented in right most panel of figure 11. The corresponding seasonal mean, sub-region averaged values of merged AODs and AAODs appear respectively in the left and middle panels of the figure 11. The error-bars represent the standard deviation, and hence the spread of the respective quantities in the spatio-temporal domain.

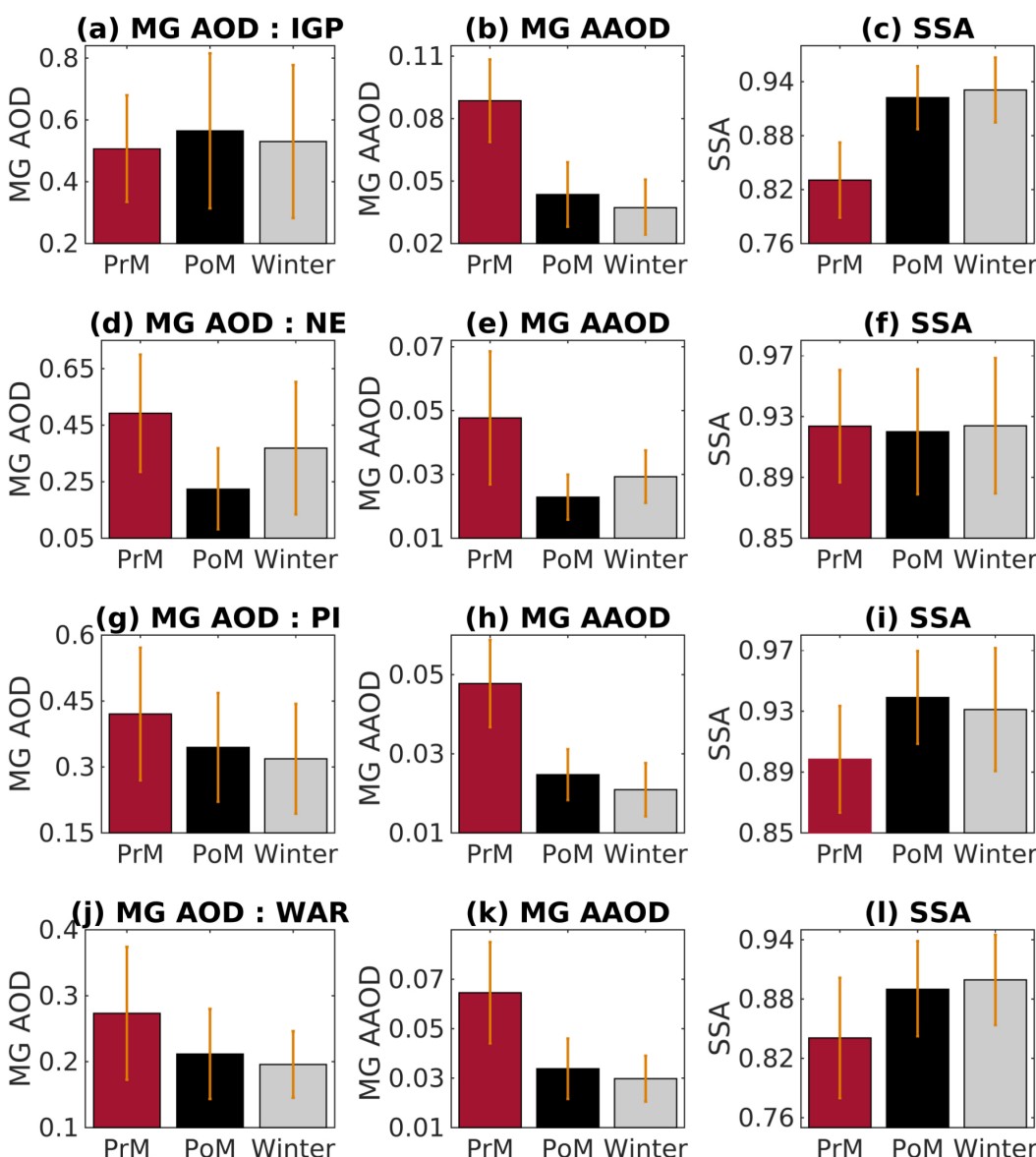

**Figure 11.** Climatological seasonal cycle of AOD (first column), AAOD (second column) and derived SSA (third column) averaged over IGP (first row), NE (second row), PI (third row) and WAR (fourth row)

The figures clearly demonstrate that amongst all the sub-regions, the highest seasonality in SSA occurs over the IGP (figure 11c), while the seasonality is lowest in the NE subregion (figure 11f). Over most of the regions, SSA is lowest in the pre-monsoon season, and highest in the winter (except for subregion PI (figure 11i)). The lower SSA over IGP which translates to increased aerosol absorption (figure 11c) during the pre-monsoon, could be largely because of transport of mineral dust from the Thar desert and west-Asian aid regions to IGP as has been reported earlier (Moorthy et al., 2005; Niranjan et al.,

2007; Beegum et al., 2008; Jethva et al., 2005). These mix with the local emissions and gets distributed vertically deep in the atmosphere (due to vigorous convective motions in the pre-monsoon season), to the regions above low-level clouds (Satheesh et al., 2008) leading to further enhancements of atmospheric absorption (Chand et al., 2009). It is to be noted that the seasonality of SSA over IGP as shown in figure 11c, is in line with Babu et al. (2016) and Vaishya et al. (2018) who have reported lesser columnar SSA during spring vis-a-vis winter over various locations (Lucknow, Ranchi, Patna and Dehradun) in IGP, based on airborne measurements of scattering and absorption coefficients.

Contrary to all the subregions, the seasonal cycle of SSA over PI shows the maxima (figure 11i) during PoM. However, SSA over WAR (figure 11l) demonstrates substantial seasonality of the similar kind as that over IGP (figure 10c), despite having considerably different seasonal variation in AOD (figure 11j) than that over IGP (figure 11a).

## 5 Summary

Gridded datasets of monthly mean aerosol optical depth and absorption aerosol optical depth have been generated for the first time over the Indian region, as a part of the SWAAMI project, by merging long-term measurements from the dense network of ground-based stations with corresponding satellite data. The merging of datasets is performed employing well-established data assimilation methods modified following a weighted interpolation scheme to account for the vertical distribution of aerosols. The gridded data demonstrated improved accuracy and conformity with independent ground-based measurements over different subregions, than the corresponding satellite datasets. The merged products also demonstrate substantially less uncertainties than those in respective satellite products, as ensured by the assimilation methodologies employed. These benefits of merged products emphasize their superiority for inputting into regional climate models. The merged AODs and AAODs reproduced the widely reported spatio-temporal features of aerosols over this region despite being significantly different (in term terms of AOD and AAOD values) from their gridded parent. The columnar SSA values have been derived from the harmonized products and their spatio-temporal variation across the domain is examined at regional and sub-regional scale. The application of these quality-enhanced, merged datasets for regional radiative forcing estimation would be discussed in Part-2 of this two-part paper.

## Appendix A: Variance in merged AODs constructed by WIM

As explained in the section 1 and 2, satellite retrieved AOD has higher uncertainties than ground-based AOD measurements, due to several reasons. As merged AOD product is developed by systematically combining SR and GE AOD by weighted interpolation method, it would be interesting to analyze and compare uncertainty in MG AOD w.r.t. that in its parent datasets.

For simplicity, we assume a case in which SR AOD at a given grid point ($X_1$) is being merged with a GE AOD ($X_2$) from ground-based aerosol observatory lying within the radius of influence from the grid point. So, following equation basic equation for WIM, we can write

$$\tilde{X} = AX_1 + BX_2 \tag{A1}$$

Here, $\tilde{X}$ is MG AOD at the given grid point and $A$ and $B$ are weights (real,positive valued scalars of size $1 \times 1$) for corresponding SR and GE AOD respectively. As, weighted interpolation method expresses MG AOD as a convex combination of SR and GE AOD, we can write

$$A + B = 1 \tag{A2}$$

which means

$$A^2 + B^2 = 1 - 2AB \tag{A3}$$

Taking variance on both sides of equation A1 we can write,

$$var(\tilde{X}) = A^2 var(X_1) + B^2 var(X_2); \tag{A4}$$

Equation A4 is expressing variance in MG AOD as a linear combination of variance in SR and GE AOD with $A^2$ and $B^2$ being respective weights. As the sum of $A^2$ and $B^2$ can never be greater than unity, as can be seen from equation A3, following inferences can be drawn regarding variance in MG AOD.

1. Variance of MG AOD can never be greater than variance of both of its parents.

2. Variance of MG AODs will always be lesser than that of SR AODs if GE AODs are available. Although depending on values of $A$ and $B$, variance of MG AOD may or may not be lesser than that of GE AOD.

3. If GE AOD is unavailable at a given location (i.e. $B = 0$) then MG AOD and its variance is exactly equal to SR AOD and its variance respectively.

These observations can be easily verified for the general case in which observations from multiple ground stations are being assimilated with the background data at a give grid point (i.e. $X_2$ is a vector).

**Appendix B:  Variance in analysed estimate by 3D-VAR**

3D-VAR constructs an analysis estimate such that the squared, weighted departures in both the parents from the analysis estimate are minimised. Here, the weights for departure in each of the parent are expressed as inverse of error covariance matrix for the corresponding parent datasets. If both parent datasets are providing unbiased estimates, the analysis estimate constructed by 3D-VAR guarantees to have minimum variance. In this section, we prove that this minimum variance for analysis estimate is guaranteed to be smaller than variances in both parent datasets.

Let $\tilde{X}$ be the unknown random variable denoting analysed (i.e. assimilated) estimate of absorption aerosol optical depth (AAOD) with mean $M$ and variance $\sigma^2$. Let $X_1$ and $X_2$ be two random variable denoting satellite retrieved AAOD (referred as SR AAOD) and ground-based AAOD (referred as GR AAOD) which are the two available, unbiased estimates of AAOD with mean $M$ and variances $\sigma_1^2$ and $\sigma_2^2$ respectively. As both SR and GR AAODs are completely independently achieved estimates, we can consider $X_1$ and $X_2$ to be uncorrelated.

The goal is to derive AAOD estimate, $\tilde{X}$ from linear combination of $X_1$ and $X_2$ such that

1. $\tilde{X}$ is a linear, unbiased estimate of AAOD, i.e. $E[\tilde{X}] = M$

2. variance of $\tilde{X}$ is minimum.

Let,

$$\tilde{X} = a_1 X_1 + a_2 X_2 \tag{B1}$$

be the assimilated estimate for AAOD where $a_1$ and $a_2$ are to be determined such that the above conditions are satisfied. Taking expectations on both sides of equation B1 we get,

$$E[\tilde{X}] = a_1 E[X_1] + a_2 E[X_2] \tag{B2}$$

$$M = M(a_1 + a_2) \quad \text{thus we get} \tag{B3}$$

$$1 = a_1 + a_2 \tag{B4}$$

Taking variance on both sides of equation B1.

$$var(\tilde{X}) = a_1^2 var(X_1) + a_2^2 var(X_2) \tag{B5}$$

We need to find $a_1$ and $a_2$ such that $var(\tilde{X})$ is minimised. Therefore, we differentiate equation B5 w.r.t. $a_1$ and equating it to zero in order to get expressions for $a_1$ and $a_2$ as,

$$a_1 = \frac{\sigma_2{}^2}{\sigma_1{}^2 + \sigma_2{}^2} \tag{B6}$$

$$a_2 = 1 - a_1 = \frac{\sigma_1{}^2}{\sigma_1{}^2 + \sigma_2{}^2} \tag{B7}$$

Substituting above expressions for $a_1$ and $a_2$ (equation B6 and B7 respectively) in equation B5, we get expression of variance in $\tilde{X}$ as,

$$var(\tilde{X}) = \left(\frac{\sigma_2{}^2}{\sigma_1{}^2 + \sigma_2{}^2}\right)^2 \sigma_1{}^2 + \left(\frac{\sigma_1{}^2}{\sigma_1{}^2 + \sigma_2{}^2}\right)^2 \sigma_2{}^2 \tag{B8}$$

The equation B8 can be further rearranged as,

$$var(\tilde{X}) = \left(\frac{\sigma_2{}^2 \sigma_1{}^2}{\sigma_2{}^2 + \sigma_1{}^2}\right) \tag{B9}$$

It can be verified that

$$\left(\frac{\sigma_2{}^2 \sigma_1{}^2}{\sigma_2{}^2 + \sigma_1{}^2}\right) \leq min\left(\sigma_1{}^2, \sigma_2{}^2\right) \tag{B10}$$

Therefore, the variance of linear, unbiased and minimum variance estimator (which is the case for 3D-VAR for the present problem) is guaranteed to be smaller that those of parent datasets. This proves that uncertainties (square root of variance) in merged AAODs constructed using 3D-VAR are guaranteed to be less than those in SR and GR AAODs.

## Appendix C: Validation of MERRA-2 PBLH

We have validated the MERRA-2 PBLH for the duration of 11 years (2008 to 2018) with those estimated using radiosonde measurements (downloaded from *http://weather.uwyo.edu/upperair/sounding.html*) over the Indian region. However, due to un-availability of continuous radiosonde measurements over many of the locations of ground-based ARFINET and/or AERONET

stations, we have considered radiosonde measurements from 8, subregional representative locations (figure 1), the Aerosol Optical Depth (AOD) and Black Carbon (BC) measurements from which are employed for constructing assimilated products. The details regarding lat-lon coordinates and broad geographical features for these stations can be found in Table S1 and S3 from the supplementary material, along with other ARFINET and AERONET stations, data from which is used for the assimilation study.

The radiosonde measurements at these stations (figure 2) are usually performed twice a day, at 00 GMT and 12 GMT and provide vertical distribution temperature, pressure, relative humidity. Further, these fundamental thermodynamic fields are used to derive the vertical profiles for virtual potential temperature ($\theta_v$), which are also provided in the respective data files.

In order to estimate PBLH from the radiosonde data, we have computed the gradient in the virtual potential temperature ($\triangle\theta_v$) at each given altitude. The height (above surface) at which the $\triangle\theta_v$ exceeds $3\,°\text{k km}^{-1}$ is considered as PBLH (Kompalli

et al., 2014; Nair et al.) at that location. The planetary boundary layer is likely to be deeper during daytime vis-a-vis nighttime, due to stronger solar heating during the day. Due to this, shallower PBL occurring in the early morning (00 GMT) may not be always captured with the provided radiosonde profiles. In the view of this, we have employed PBLH estimated using radiosonde measurements during daytime (12 GMT) only, for the present validation purposes.

The hourly averaged PBLH (12 GMT) given by MEERA-2 for that particular day, are bi-linearly interpolated to the locations

of stations shown in figure 2, in order to get spatio-temporally collocated estimate of MERRA-2 PBLH. The scatter plots between the collocated PBLH and those estimated from radiosonde measurements for 8 locations, during year 2008 to 2018, are presented in figure 3.

It can be seen from figure 3 that, PBLH provided by MERRA-2 dataset are well-correlated with those estimated using radiosonde data, although the correlation coefficient is varying from 0.63 to 0.96, w.r.t the location. The equations for linear

regression between the two PBLH estimates suggest that, PBLH given by MERRA-2 are underestimated over majority of the stations (figure 3a to 3e), which is in line with the general observation made by reviewer. Nonetheless, substantially overestimated PBLH values by MERRA-2 are apparent for some of the stations (figure 3f to 3h).

*Author contributions.*

HSP carried out analyzing, modifying and finalizing the assimilation methods and further employed them for assimilating the

ground-based measurements of AOD and BC with respective satellite retrieved products. HSP was also primarily responsible for writing the manuscript which is further reviewed and edited by KKM, SKS, RSN and SL. SKS, SSB and KKM provided the ARFINET (ground-based) data which is critically important for carrying out the present work. The valuable guidance

regarding aerosols was given by SKS and KKM, while the valuable inputs regarding the data assimilation methodologies were provided by SL, RSN and KKM.

*Competing interests.* The authors declare that they have no conflict of interest.

*Acknowledgements.* This work is carried out as a part of the project titled "South West Asian Aerosol Monsoon Interactions (SWAAMI)
5   (Grant No:MM/NERC-MoES- 1/2014/002)"funded by the Ministry of Earth Sciences (MoES), New Delhi. We thank all the ARFINET investigators for the continuous efforts and support provided in maintaining the network as well as in collecting and processing the data. We thank the AERONET (data available at *http://aeronet/gsfc.nasa.gov)* PIs and their staff for establishing and maintaining the sites used in this investigation. The Terra, Aqua MODIS Aerosol Optical Depth Monthly, L3, Global, 1 Deg. CMG data sets were acquired from the Level-1 and Atmosphere Archive and Distribution System (LAADS) Distributed Active Archive Center (DAAC), located in the Goddard
10   Space Flight Center in Greenbelt, Maryland *(https://ladsweb.nascom.nasa.gov/).* MISR Aerosol Optical Depth, Monthly L3 Global data were obtained from the NASA Langley Research Center Atmospheric Science Data Center. We also acknowledge Global Modeling and Merging Office (GMAO) and the GES DISC for distribution of MERRA data. We would like take this opportunity to thank Dr. Ashwin Seshadri for his valuable suggestions during the work. We also thank Dr. Hiren Jethva for providing the OMI data.

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
