# Peer review of "Assessment of Regional Aerosol Radiative Effects under SWAAMI Campaign – PART 1: Quality-enhanced Estimation of Columnar Aerosol Extinction and Absorption Over the Indian Subcontinent"

_Atmospheric Chemistry and Physics, 2019_

## Referee Comment (RC1) · Anonymous Referee #1 · 18 Apr 2019

For such comparative estimation a higher resolution data from satellites such as level 2 data available almost daily be ideal, while level 3 data at a lower resolution on a monthly basis has been reported to be systematically underestimated.

Comparison between AODs and AAODs are expressed in terms of R, Rˆ2 has used to explain such variation. R values lower.

dAOD and dAAOD values are large, sometimes as equal to AOD or AAOD.

[Figure]

The uncertainties are reported to be lower than satellite product, how and why this happens is not clear.

Using BC mass and dust to construct SSA can result in uncertainty, for example, the aethalometer measurements are reported to have uncertainties in BC mass measured at different environmental conditions using the same attenuation coefficinets valid for urban regions.

As for as the methodology is concerned several approcahes are mentioned with variations in planetary boundary layer height etc., sensitivity of these assumptions not explained.

---

## Referee Comment (RC2) · Anonymous Referee #2 · 29 Apr 2019

Review on 'Assessment of Regional Aerosol Radiative Effects under SWAAMI Campaign – PART 1: Quality-enhanced Estimation of Columnar Aerosol Extinction and Absorption Over the Indian Subcontinent' by Harshavardhana Sunil Pathak et al., (ACP-2019-153)

This manuscript presents the construction of quality enhanced aerosol optical depth (AOD) and absorption aerosol optical depth (AAOD) while suitably assimilating the observations from the ground based network (ARFINET and AERONET) and satellite (MODIS, MSIR, Kalpana-1, INSAT-3A, OMI) derived products. As rightly mentioned, ground based observations are more accurate and have good temporal sampling, though spatial coverage is poor than their counter parts (satellite measurements). Authors have taken full advantage of their (ground based and space borne) respective limitations and could able to produce more accurate gridded observations by suitably merging various data sets. Final products obtained through assimilation have been further validated with independent observations and could notice improved correlations, particularly in AODs. Finally, aerosol properties in different months representing contrasting seasons (winter and summer) has been presented in this part of the manuscript while leaving detailed results and discussion in part 2.

In general, results presented in this manuscript are unique which are first of its kind and authors made nice compilation of different data sets over Indian region where temporal and spatial heterogeneity of aerosol properties are large when compared to the rest of the world.

In general, paper is well written and will be interest to the researchers working in the aerosol field and very apt for publishing in ACP. However, there are few issues and sometimes interpretation is missing at some instances which demands careful editing or re-writing. Below are the some of the issues which authors may consider while revising the manuscript. Authors are strongly encouraged to revise and re-submit this manuscript.

Specific major comments/suggestions:

1. Page 8: Line 3. It was mentioned that PBL height is used from MERRA-2 and it has been validated previously by Sathyanadh et al. (2017). Though Sathyanadh et al. (2017) mentioned that good correlation between 0.74-0.83 is seen when MERRA-2 PBLH when compared with radiosonde and radio occultation (done for very few stations that too for one year 2011 only), our experience is that it underestimates heavily

the PBLH. Since this is the one of important parameter while calculating AAOD, it is suggested to show detailed comparison of MERRA-2 with existing IMD radiosonde derived PBL heights or GPS RO measurements for the said period 2008-2016. A figure showing the PBL altitudes for the respective stations will be highly useful while interpreting the results particularly the AAODs.

2. Validation with independent measurements of AAOD: I am surprised to very poor correlations in AAOD shown in Figure 3. Since correlations are poor, how to trust the data for further applications. Perhaps need to be re-checked while using actual PBL heights.

3. I suggest adding another panel in Figures 4-5 showing the difference between SR AOD and MG AOD along with dAOD. I do not understand why the difference is not shown throughout the Indian region similar to that shown for dAAOD in Figures 6-7.

4. In the abstract it is listed as 44 stations for AOD and 32 stations for AAOD. However, I am unable to see them in the list of stations provided in the supplementary information.

Minor comments/suggestions:

1. Page 2: Line 21: remove repeated word 'have'

2. Figure 1: dot size used in this figure is too small to recognize different colors. Size should increase up to 3-4 times similar to that shown in Figures 4-7.

3. I suggest moving Figure 9 to supplementary information as the regional coordinates are already mentioned in Table 1. This figure is not adding much.
* * *

---

## Author Comment (AC1) · 9 Jun 2019

**Responses to comments from Anonymous Referee #1**

June 10, 2019

We are thankful towards both the reviewers for their constructive comments and suggestions which not only provided the deeper insights into the present work but also helped to improve the overall quality of the manuscript. The responses to the specific comments/suggestions from the Anonymous Referee #1 are as follows. The referee comments are shown in red color and our responses are shown in black color.

**1 Answers to comments from Anonymous Referee #1**

**Comment 1**: For such comparative estimation a higher resolution data from satellites such as level 2 data available almost daily be ideal, while level 3 data at a lower resolution on a monthly basis has been reported to be systematically underestimated.

We agree with reviewer that level-2, daily data would be more appropriate for construction of assimilated product. However, level-2, daily satellite data often consists of large data gaps, primarily due to clouds. This issue is even more prominent in case of retrievals using space-borne measurements by Ozone Monitoring Instrument (OMI) which has coarser resolution (13 km × 25 km) than that of MODIS. In addition, the ground-based AOD measurements from ARFINET can be performed only in clear-sky conditions, which limits the availability of ground-based AODs at daily time scale.

In order to improve the regional climate impact assessment for aerosols, one needs to construct quality improved, spatially homogeneous and temporally continuous datasets for aerosol properties, which is also envisioned in South West Asian Aerosol Monsoon Interactions (SWAAMI), a joint Indo-UK field experiment. In accordance with this, we have employed level-3, monthly satellite products, as background datasets from which wide and continuous spatial coverage is inherited by assimilated products.

**Comment 2**: Comparison between AODs and AAODs are expressed in terms of R, R2 has used to explain such variation. R values lower

We agree with reviewers observation about the correlation between merged AAODs and independent ground-based AAODs (R) being low yet significant (at 95% confidence level). However, there are genuine reasons behind the same which are as explained below.

The merged AAOD (MG AAOD) product is developed by systematically assimilating Ozone Monitoring Instrument (OMI) retrieved AAODs and those estimated from ground-based BC measurements as well as satellite-based infrared measurements, employing 3D-VAR, a widely used assimilation technique based on weighted least square error minimization. The OMI AAODs which form the background data for the AAOD assimilation are demonstrating relatively weaker yet significant correlation (R=0.36) with independent ground-based AAODs (GR AAOD) (Page no.16, figure 3a). Such weak correlation between OMI and GR AAODs could be primarily because of differences in the estimation procedures. OMI-near UV (OMAERUV) algorithm which is employed for retrieval of AOD and AAOD, makes use of the measurements of the upwelling radiation at 354 and 388 nm at the top of atmosphere (TOA). This algorithm exploits the prominent interaction between molecular scattering and the aerosol absorption as well as lower surface reflectance in UV wavelength range. The AOD and AAODs are further retrieved using the look up tables (LUT) consisting of pre-computed reflectance values (at the TOA) derived by a set of aerosol models which consider specific vertical distribution for each of

the aerosol types (Torres et al., 2005, 2007). For carbonaceous aerosols, those models consider exponential profile with maximum concentration occurring at 3 km above ground level. On the other hand, during estimation of AAODs corresponding to surface level BC mass concentration, we have considered uniform distribution of BC within the PBL and exponential decay above it. Being based on the common inferences drawn from the extensive aircraft and balloon measurements of BC over different regions of India (Suresh Babu et al., 2010; Babu et al., 2011), the vertical distribution considered in our work is better representative for the Indian region. This difference between vertical distribution of aerosols considered during estimation of OMI and GR AAOD could have lead to weak correlation between the two. In addition, the uncertainties in OMI AAODs emanating from assumptions about height of an aerosol layer and sub-pixel cloud contamination could have further assisted in reducing the correlation between OMI AAOD and their ground-based counterpart. This weak correlation between OMI and GR AAODs could be one of the primary reasons behind the observed correlation between merged and GR AAODs (R = 0.47, figure 3b, page no 16 from the earlier version of manuscript). As suggested by reviewer, $R^2$ values are also now included in figure 2 and 3 from the earlier version of manuscript,

As described by 3D-VAR, the weights given to each parent dataset are inversely proportional to the uncertainties in the respective datasets (Kalnay, 2003; Lewis et al., 2006). Accordingly, GR AAODs are inversely weighted with respective uncertainties (specified in equation 14, page no. 15 of earlier version of manuscript), which certainly limits the signature of ground-based measurements in assimilated AAODs. This factor also could have further restricted the correlation between merged AAODs with its ground-based counterpart. In spite of this, the point to be highlighted here is that the correlation shown by merged AAOD with independent GR AAOD (R = 0.47, figure 3b, page no 16 of earlier form of manuscript) is about 30% higher than that shown by OMI AAOD (R = 0.36, figure 3a, page no 16 of the previous version of manuscript),

which underlines the substantial improvement brought in due to assimilation.

**Comment 3** dAOD and dAAOD values are large, sometimes as equal to AOD or AAOD

We agree with this observation made by reviewer that dAOD (i.e. MG AOD - SR AOD) and dAAOD (i.e. MG AAOD - SR AAOD) values are sometimes as large as AOD and AAODs shown by respective satellite products.

The merged AOD and AAODs are constructed by systematically assimilating GR AOD and AAODs with corresponding satellite products employing data assimilation schemes as explained in section 3.1. Therefore the dAOD and dAAOD values are primarily associated with differences between corresponding satellite retrievals and ground-based measurements. The MODIS retrieved AODs tend to be underestimated as compared to AERONET AODs over highly polluted, smoke covered regions (Zhang and Reid, 2006). However, cloud contamination leads to systematic overestimation of MODIS AODs irrespective of AOD ranges (Zhang and Reid, 2006). This issue is even more prominent in case of OMI AAODs which tend to be overestimated due to sub-pixel cloud contamination (Torres et al., 2005, 2007). In addition, the assumptions regarding vertical distribution of absorbing aerosols made by the aerosol models used in OMI-near UV (OMAERUV) algorithm, can also lead to OMI AAODs being substantially different from their ground-based counterpart. The difference between satellite retrieved and ground-measured values, is the main reason behind the kind of dAOD and dAAOD values shown in our work.

However, the validation exercise (section 4.1) has demonstrated that the assimilated products are better confirming with independent ground-based measurements than their satellite counterparts, which indicates the improved accuracy of assimilated products vis-a-vis respective satellite datasets.

**Comment 4** The uncertainties are reported to be lower than satellite product, how and why this happens is not clear.

We agree that the explanation about the uncertainty reduction for assimilated products is not directly given in the earlier version of the manuscript.

We would like draw referee's kind attention towards that fact that uncertainties in assimilated AOD and AAODs are guaranteed to be lower than those in respective satellite products, as guaranteed by the assimilation techniques employed, Weighted Interpolation Method (WIM, used for AOD assimilation) and 3D-VAR (used for AAOD assimilation). Theoretical proof for property of variance minimization for WIM is given in section S2 in the supplementary material (earlier version). Although we did not provide the similar proof for 3D-VAR, the appropriate references (Kalnay, 2003; Lewis et al., 2006) for the same were given in the earlier form of manuscript (line no 13-15, page no. 15). Nonetheless, in the modified version of the manuscript, we are including the the proofs for variance minimization property demonstrated by WIM and 3D-VAR in section 3.

**Comment 5** Using BC mass and dust to construct SSA can result in uncertainty, for example, the aethalometer measurements are reported to have uncertainties in BC mass measured at different environmental conditions using the same attenuation coefficients valid for urban regions.

We agree with this important point raised by reviewer about sources of uncertainties in SSA emanating from those in measurements of BC mass and dust. The uncertainties in BC mass concentration measurements made by Aethalometer, are reported to be 2 to 5 % (Hansen and Novakov, 1990; Babu et al., 2004; Dumka et al., 2010). In the present work, we have considered 5% uncertainties in BC mass concentration measurements while estimating the uncertainties in BC AAODs, as mentioned in line 30, page no. 14 of the earlier version of manuscript. The details regarding estimation of uncertainties in BC AAODs are provided on line no. 26-30 on page no. 14 and line no. 1-3 on page no.15 of the earlier form of manuscript. Similarly, the uncertainties in dust AAODs emanating from vertical heterogeneities in dust and its optical properties are estimated to be 25%, as mentioned on line no.4, page no. 15. The composite variance in GR AAODs (i.e. BC AAOD + Dust AAOD) estimated by equation 14 (as given on page no. 15 of earlier version of manuscript) was used to form

diagonal elements of observation error covariance matrix (R) used in 3D-VAR (equation 13 in earlier form of manuscript).

Thus, the uncertainties in BC mass measurements and dust, were taken into account while constructing assimilated AAODs which are further used along with assimilated AODs to estimate SSA (equation 17 on page no. 22 in earlier form of manuscript). The uncertainties in SSA are then estimated from those in MG AODs and AAODs, as described in line no. 13-15, page no. 23 from the earlier version of manuscript. Thus, the uncertainties contributed by BC and dust AAODs are being taken into account to estimate those in SSA.

**Comment 6** As far as the methodology is concerned several approaches are mentioned with variations in planetary boundary layer height etc., sensitivity of these assumptions not explained.

We understand the concern raised by the reviewer regarding robustness of assumptions which is usually examined through sensitivity analysis. However, we would like note that, variance in planetary boundary layer height (PBLH) data is not assumed, rather estimated using long-term time series (year 2000-2013) of PBLH data provided by MERRA-2 reanalysis product. In order to do so, we have constructed error covariance matrix using monthly mean, MERRA-2 PBLH after removing long-term trend (if existing) and seasonal variation. This covariance matrix is constructed employing the methodology explained in section S3.1, page no. S6 from the earlier version of supplementary material. The diagonal terms of the covariance matrix provide estimate for the variance in PBLH for the corresponding grid points, which we have considered in the subsequent calculations. Therefore, we would like to kindly note that the prescribed sensitivity analysis is not warranted.

We understand that this was not explained in the earlier version of manuscript due to which it was seeming that the variance in PBLH is assumed, which is not the case. In the view of this, we are including these details in the modified version of supplementary material.

**References**

Babu, S. S., Moorthy, K. K., and Satheesh, S.: Aerosol black carbon over Arabian Sea during intermonsoon and summer monsoon seasons, Geophysical Research Letters, 31, 2004.

Babu, S. S., Moorthy, K. K., Manchanda, R. K., Sinha, P. R., Satheesh, S., Vajja, D. P., Srinivasan, S., and Kumar, V.: Free tropospheric black carbon aerosol measurements using high altitude balloon: do BC layers build "their own homes" up in the atmosphere?, Geophysical research letters, 38, 2011.

Dumka, U., Moorthy, K. K., Kumar, R., Hegde, P., Sagar, R., Pant, P., Singh, N., and Babu, S. S.: Characteristics of aerosol black carbon mass concentration over a high altitude location in the Central Himalayas from multi-year measurements, Atmospheric Research, 96, 510 – 521, doi:https://doi.org/10.1016/j.atmosres.2009.12.010, URL http://www.sciencedirect.com/science/article/pii/S0169809509003494, 2010.

Hansen, A. and Novakov, T.: Real-time measurement of aerosol black carbon during the carbonaceous species methods comparison study, Aerosol Science and Technology, 12, 194–199, 1990.

Kalnay, E.: Atmospheric modeling, data assimilation and predictability, Cambridge university press, 2003.

Lewis, J. M., Lakshmivarahan, S., and Dhall, S.: Dynamic data assimilation: a least squares approach, vol. 104, Encyclopedia of Mathematics and its Applications, Cambridge University Press, 2006.

Suresh Babu, S., Krishna Moorthy, K., and Satheesh, S.: Vertical and horizontal gradients in aerosol black carbon and its mass fraction to composite aerosols over the east coast of Peninsular India from Aircraft measurements, Advances in Meteorology, 2010, 2010.

Torres, O., Bhartia, P. K., Sinyuk, A., Welton, E. J., and Holben, B.: Total Ozone Mapping Spectrometer measurements of aerosol absorption from space: Comparison to SAFARI 2000 groundbased observations, Journal of Geophysical Research: Atmospheres, 110, doi:10.1029/2004JD004611, URL https://agupubs.onlinelibrary.wiley.com/doi/abs/10.1029/2004JD004611, 2005.

Torres, O., Tanskanen, A., Veihelmann, B., Ahn, C., Braak, R., Bhartia, P. K., Veefkind, P., and Levelt, P.: Aerosols and surface UV products from Ozone Monitoring Instrument observations: An overview, Journal of Geophysical Research: Atmospheres, 112, doi:10.1029/2007JD008809, URL https://agupubs.onlinelibrary.wiley.com/doi/abs/10.1029/2007JD008809, 2007.

Zhang, J. and Reid, J. S.: MODIS aerosol product analysis for data assimilation: Assessment of over-ocean level 2 aerosol optical thickness retrievals, Journal of Geophysical Research: Atmospheres, 111, n/a–n/a, doi: 10.1029/2005JD006898, URL http://dx.doi.org/10.1029/2005JD006898, d22207, 2006.

---

## Author Comment (AC2) · 9 Jun 2019

**Responses to comments from Anonymous Referee #2**

June 10, 2019

We are thankful towards both the reviewers for their constructive comments and suggestions which not only provided the deeper insights into the present work but also helped to improve the overall quality of the manuscript. The responses to the specific comments/suggestions from the Anonymous Referee #2 are as follows. The referee comments are shown in red color and our responses are shown in black color.

**1 Answers to major comments from Anonymous Referee #2**

**Comment 1**: Page 8: Line 3. It was mentioned that PBL height is used from MERRA-2 and it has been validated previously by Sathyanadh et al. (2017). Though Sathyanadh et al. (2017) mentioned that good correlation between 0.74-0.83 is seen when MERRA-2 PBLH when compared with radiosonde and radio occultation (done for very few stations that too for one year 2011 only), our experience is that it underestimates heavily the PBLH. Since this is the one of important parameter while calculating AAOD, it is suggested to show detailed comparison of MERRA-2 with existing IMD radiosonde derived PBL heights or GPS RO measurements for the said period 2008-2016. A figure showing the PBL altitudes for the respective stations will be highly useful while interpreting the results particularly the AAODs.

We are thankful towards the reviewer for correctly pointing out that the

validation of planetary boundary layer height (PBLH) by MERRA-2 dataset, provided in Sathyanadh et al. (2017) is limited in terms of duration (May to September 2011). Therefore, in the view of importance of accurate PBLH for our studies, it is pertinent to validate PBLH provided by MERRA-2 over the Indian region for the entire duration. Accordingly, we have validated the MERRA-2 PBLH for the duration of 11 years (2008 to 2018) with those estimated using radiosonde measurements (downloaded from *http://weather.uwyo.edu/upperair/sounding.html*) over the Indian region. However, due to unavailability of continuous radiosonde measurements over many of the locations of ground-based ARFINET and/or AERONET stations, we have considered radiosonde measurements from 8, subregional representative locations (Figure 1), the Aerosol Optical Depth (AOD) and Black Carbon (BC) measurements from which are employed for constructing assimilated products. The details regarding lat-lon coordinates and broad geographical features for these stations are provided in the Table S1 and S3 from the supplementary material, along with other ARFINET and AERONET stations, data from which is used for the assimilation study.

[Figure]

Figure 1: Locations of the ground stations, radiosonde measurements from which are used for the purpose of validating PBLH derived by MERRA-2. These subregional representative stations form a subset of ground-based observatories, AOD and BC mass concentration measurements from which are employed for construction of assimilated AOD and Absorption AOD (AAOD) products.

The radiosonde measurements at these stations (figure 1) are usually performed twice a day, at 00 GMT and 12 GMT and provide vertical distribution temperature, pressure, relative humidity. Further, these fundamental thermodynamic fields are used to derive the vertical profiles for virtual potential temperature ($\theta_v$), which are also provided in the respective data files.

In order to estimate PBLH from the radiosonde data, we have computed the gradient in the virtual potential temperature ($\triangle\theta_v$) at each given altitude. The height (above surface) at which the $\triangle\theta_v$ exceeds 3 °k km$^{-1}$ is considered as PBLH (Kompalli et al., 2014; Nair et al., 2011) at that location. The planetary boundary layer is likely to be deeper during daytime vis-a-vis nighttime, due to stronger solar heating during the day. Due to this, shallower PBL occurring in the early morning (00 GMT) may not be always captured with the provided radiosonde profiles. In the view of this, we have employed PBLH estimated using radiosonde measurements during daytime (12 GMT) only, for the present validation purposes.

The hourly averaged PBLH (12 GMT) given by MEERA-2 for that particular day, are bi-linearly interpolated to the locations of stations shown in figure 1, in order to get spatio-temporally collocated estimate of MERRA-2 PBLH. The scatter plots between the collocated PBLH and those estimated from radiosonde measurements for 8 locations, during year 2008 to 2018, are presented in figure 2.

[Figure]

Figure 2: Comparison of spatio-temporally collocated MERRA-2 PBLH with those derived from radiosonde measurements performed at 8 representative locations during year 2008 to 2018. The correlation coefficient (R) (significant at 95% confidence limit) and the equation of linear regression between the two PBLH estimates are provided in each of the figures.

It can be seen from figure 2 that, PBLH provided by MERRA-2 dataset are well-correlated with those estimated using radiosonde data, although the correlation coefficient is varying from 0.63 to 0.96, w.r.t the location. The equations for linear regression between the two PBLH estimates suggest that, PBLH given by MERRA-2 are underestimated over majority of the stations (Figure 2a to 2e), which is in line with the general observation made by reviewer. Nonetheless, substantially overestimated PBLH values by MERRA-2 are apparent for some of the stations (Figure 2f to 2h).

We agree with the reviewer that, in order to enhance the accuracy of AAODs estimated from ground-based BC mass concentration measurements, one would use PBLH values derived from radiosonde data. However, due to limited temporal sampling (daily 2 profiles only) and unavailability of radiosonde measurements at every location of ARFINET observatory (34 in number), we had to rely on the PBLH product provided by MERRA-2 reanalysis dataset, which is well correlated with observations. As suggested by the reviewer, we are adding figure 2 and its pertinent explanation in the modified version of manuscript.

**Comment 2**: Validation with independent measurements of AAOD: I am surprised to very poor correlations in AAOD shown in Figure 3. Since correlations are poor, how to trust the data for further applications. Perhaps need to be re-checked while using actual PBL heights.

We agree with the observation made by referee about the correlation between merged AAODs and independent ground-based AAODs which is slightly low yet significant (at 95% confidence level). However, there are genuine reasons behind the same which are as explained below.

The merged AAOD (MG AAOD) product is developed by systematically assimilating Ozone Monitoring Instrument (OMI) retrieved AAODs and those estimated from ground-based BC measurements as well as satellite-based infrared measurements, employing 3D-VAR, a widely used assimilation technique based on weighted least square error minimization. The OMI AAODs which form the background data for the AAOD assimilation are demonstrating relatively weaker yet significant correlation (R=0.36) with independent ground-based AAODs (GR AAOD) (Page no.16, figure 3a). Such weak correlation between OMI and GR AAODs could be primarily because of differences in the estimation procedures. OMI-near UV (OMAERUV) algorithm which is employed for retrieval of AOD and AAOD, makes use of the measurements of the upwelling radiation at 354 and 388 nm at the top of atmosphere (TOA). This algorithm exploits the prominent interaction between molecular scattering and the aerosol absorption as well as lower surface reflectance in UV wavelength range. The AOD and AAODs are further retrieved using the look up tables (LUT) consisting of pre-computed reflectance values (at the TOA) derived by a set of aerosol models which consider specific vertical distribution for each of the aerosol types (Torres et al., 2005, 2007). For carbonaceous aerosols, those models consider exponential profile with maximum concentration occurring at 3 km above ground level. On the other hand, during estimation of AAODs corresponding to surface level BC mass concentration, we have considered uniform distribution of BC within the PBL and exponential decay above it. Being based on the common inferences drawn from the extensive aircraft and balloon measurements of BC over different regions of India (Suresh Babu et al., 2010; Babu et al., 2011), the vertical distribution considered in our work is better representative for the Indian region. This difference between vertical distribution of aerosols considered during estimation of OMI and GR AAOD could have lead to weak correlation between the two. In addition, the uncertainties in OMI AAODs emanating from assumptions about height of an aerosol layer and sub-pixel cloud contamination could have further assisted in reducing the correlation between OMI AAOD and their ground-based counterpart. This weak correlation between OMI and GR AAODs could be one of the primary reasons behind the observed correlation between merged and GR AAODs (R = 0.47, figure 3b, page no 16 from the earlier version of manuscript)

As described by 3D-VAR, the weights given to each parent dataset are inversely proportional to the uncertainties in the respective datasets (Kalnay,

2003; Lewis et al., 2006). Accordingly, GR AAODs are inversely weighted with respective uncertainties (specified in equation 14, page no. 15 of earlier version of manuscript), which certainly limits the signature of ground-based measurements in assimilated AAODs. This factor also could have further restricted the correlation between merged AAOD with its ground-based counterpart. In spite of this, the point to be highlighted here is that the correlation shown by merged AAOD with independent GR AAOD (R = 0.47, figure 3b, page no 16 of earlier form of manuscript) is about 30% higher than that shown by OMI AAOD (R = 0.36, figure 3a, page no 16 from the previous version of manuscript), which underlines the substantial improvement brought in due to assimilation.

**Comment 3**: I suggest adding another panel in Figures 4-5 showing the difference between SR AOD and MG AOD along with dAOD. I do not understand why the difference is not shown throughout the Indian region similar to that shown for dAAOD in Figures 6-7.

The difference between the MG and SR AOD which is indicated by dAOD (i.e MG AOD - SR AOD) is shown in Figure 4c and 5c (page no. 18 and 19 from earlier version of manuscript). Similarly, spatial variation of dAAOD (i.e. MG AAOD - SR AAOD) is demonstrated in Figure 6c and 7c (page no. 18 and 19 from earlier version of manuscript), for the two representative cases. As can be seen from Figure 4c and 5c that non-zero dAOD values are being demonstrated over the regions represented by ground-based AODs and they (dAOD values) smoothly reduce to zero as one moves away from the locations of ground-based observatories. On the other hand, Figure 6c and 7c are showing dAAODs having wider spatial coverage The differences in the nature of regional distribution for dAOD (Figure 4c and 5c) and dAAOD (Figure 6c and 6c) are primarily due to difference between nature of assimilation methods employed for AOD and AAOD.

As detailed in section 3.1 (earlier version of manuscript), for AOD assimilation, we have employed Weighted Interpolation Method (WIM) (a variation to the Successive Correction Method) which ensures that MG AODs are always

bounded by SR and GR AODs and are guaranteed to have less uncertainties than those in SR AODs (section S2 from the earlier version of supplementary material). WIM provides a localized approach towards assimilation and merges the GR AODs with SR AODs at the grid points within specified radius and height of influence. Nevertheless, for AAOD assimilation, WIM could not smoothly merge GR AAODs with the gridded background data (i.e. SR AAODs), possibly due to the relatively weaker correlation between the two (R = 0.36, Figure 3a from the earlier version of manuscript) vis-a-vis GR AOD and SR AOD (R = 0.77, Figure 2a from the earlier version of manuscript). In the view of this, we have employed, a widely used data assimilation technique, 3D-VAR, which merges the scattered observations with the gridded data in the patterns dictated by the background error covariance matrix (B). In the present work, B is constructed employing long-term time series (year 2005 to 2016) for the monthly OMI AAODs over the Indian region (section S3 from the earlier version of supplementary material). Due to this, the differences between MG and SR AAODs are being seen over entire Indian region (Figure 6c and 7c), unlike in case of AOD where the differences between the merged and satellite product are prominent over the regions represented by ground-based AODs.

**Comment 4**: In the abstract it is listed as 44 stations for AOD and 32 stations for AAOD. However, I am unable to see them in the list of stations provided in the supplementary information.

Table S1 and S2 from the supplementary material (previous version) provide the lists of 27 ARFINET and 20 AERONET stations, AOD measurements from which are employed for assimilation purpose. As three locations are common among ARFINET and AERONET, the total tally of stations reduces to 44. The list of 34 ARFINET stations providing BC mass concentration measurements, is given in Table S3 from earlier version of the supplementary material.

**2 Answers to minor comments from Anonymous Referee #2**

**Comment 1** Page 2: Line 21: remove repeated word have

This rectification has been incorporated in the modified manuscript.

**Comment 2** Figure 1: dot size used in this figure is too small to recognize different colors. Size should increase up to 3-4 times similar to that shown in Figures 4-7

We have modified the Figure 1 demonstrating locations of all ground-based stations, data from which is used in the current work. The modified form of Figure 1 is incorporated in the updated version of manuscript.

**Comment 3** I suggest moving Figure 9 to supplementary information as the regional coordinates are already mentioned in Table 1. This figure is not adding much.

The suggested change has been implemented in the modified form of manuscript.

**References**

Babu, S. S., Moorthy, K. K., Manchanda, R. K., Sinha, P. R., Satheesh, S., Vajja, D. P., Srinivasan, S., and Kumar, V.: Free tropospheric black carbon aerosol measurements using high altitude balloon: do BC layers build "their own homes" up in the atmosphere?, Geophysical research letters, 38, 2011.

Kalnay, E.: Atmospheric modeling, data assimilation and predictability, Cambridge university press, 2003.

Kompalli, S. K., Babu, S. S., Moorthy, K. K., Manoj, M., Kumar, N. K., Shaeb, K. H. B., and Joshi, A. K.: Aerosol black carbon characteristics over Central India: Temporal variation and its dependence on mixed layer height, Atmospheric research, 147, 27–37, 2014.

Lewis, J. M., Lakshmivarahan, S., and Dhall, S.: Dynamic data assimilation: a

least squares approach, vol. 104, Encyclopedia of Mathematics and its Applications, Cambridge University Press, 2006.

Nair, S. K., Anurose, T., Subrahamanyam, D. B., Kumar, N., Santosh, M., Sijikumar, S., Mohan, M., and Namboodiri, K.: Characterization of the Vertical Structure of Coastal Atmospheric Boundary Layer over Thumba (8.5 N, 76.9 E) during Different Seasons., Advances in Meteorology, 2011, 2011.

Sathyanadh, A., Prabhakaran, T., Patil, C., and Karipot, A.: Planetary boundary layer height over the Indian subcontinent: Variability and controls with respect to monsoon, Atmospheric Research, 195, 44 – 61, doi:https://doi.org/10.1016/j.atmosres.2017.05.010, URL `http://www.sciencedirect.com/science/article/pii/S0169809517305549`, 2017.

Suresh Babu, S., Krishna Moorthy, K., and Satheesh, S.: Vertical and horizontal gradients in aerosol black carbon and its mass fraction to composite aerosols over the east coast of Peninsular India from Aircraft measurements, Advances in Meteorology, 2010, 2010.

Torres, O., Bhartia, P. K., Sinyuk, A., Welton, E. J., and Holben, B.: Total Ozone Mapping Spectrometer measurements of aerosol absorption from space: Comparison to SAFARI 2000 groundbased observations, Journal of Geophysical Research: Atmospheres, 110, doi:10.1029/2004JD004611, URL `https://agupubs.onlinelibrary.wiley.com/doi/abs/10.1029/2004JD004611`, 2005.

Torres, O., Tanskanen, A., Veihelmann, B., Ahn, C., Braak, R., Bhartia, P. K., Veefkind, P., and Levelt, P.: Aerosols and surface UV products from Ozone Monitoring Instrument observations: An overview, Journal of Geophysical Research: Atmospheres, 112, doi:10.1029/2007JD008809, URL `https://agupubs.onlinelibrary.wiley.com/doi/abs/10.1029/2007JD008809`, 2007.